

# Generation of super-resolution gap-free ocean colour satellite products using DINEOF.

Aida Alvera-Azcárate[1], Dimitry Van der Zande[2], Alexander Barth[1], Antoine Dille[2], Joppe Massant[2], and Jean-Marie Beckers[1]

[1]AGO-GHER, University of Liège, Allée du Six Aout, 17, Sart Tilman, Liège 4000, Belgium
[2]Royal Belgian Institute of Natural Sciences (RBINS), Direction Natural Environment Rue Vautier 29, 1000 Brussels, Belgium

**Correspondence:** Aida Alvera-Azcárate (a.alvera@uliege.be)

**Abstract.** In this work we present a super-resolution approach to derive high spatial and temporal resolution ocean colour satellite datasets. The technique is based on DINEOF (Data Interpolating Empirircal Orthogonal Functions), a data-driven method that uses the spatio-temporal coherence of the analysed datasets to infer missing information. DINEOF is now used to effectively increase the spatial resolution of satellite data, and is applied to a combination of Sentinel-2 and Sentinel-3 datasets.
The results show that DINEOF is able to infer the spatial variability observed in the Sentinel-2 data into the Sentinel-3 data, while reconstructing missing information due to clouds and reducing the amount of noise in the initial dataset. In order to achieve this, both Sentinel-2 and Sentinel-3 datasets have undergo the same preprocessing, including a comprehensive, region-independent, and pixel-based automatic switching scheme for choosing the most appropriate atmospheric correction and ocean colour algorithm to derive the in-water products. The super-resolution DINEOF has been applied to two different variables
(turbidity and chlorophyll) and two different domains (Belgian coastal zone and the whole North Sea), and the submesoscale variability of the turbidity along the Belgian coastal zone has been studied.

## 1 Introduction

The coastal ocean is a very dynamic region, both in space and time. Coastal regions are subject to strong anthropogenic pres-
sure, and satellite data provide the necessary spatial and temporal coverage to study and monitor these regions. There is a need however to measure these areas at both high spatial and temporal resolution, in order to capture the relevant scales of variability. While "traditional" ocean colour satellites like Sentinel-3 provide daily temporal resolution, the sensors onboard these satellites do not measure at the necessary high spatial resolution to resolve complex coastal dynamics. High spatial resolution sensors, like the MultiSpectral Instrument (MSI) onboard Sentinel-2 (10m-60m resolution), are able to resolve these
small scales, but their temporal revisit time is far from optimal (about 5 days considering the Sentinel-2 A and B tandem). Additionally, both high spatial resolution datasets and traditional ones are hindered by the presence of clouds, resulting in a large amount of missing data.





Super-resolution approaches aimed at increasing the spatial resolution of geophysical datasets have been developed using
neural network methodologies. For example, Thiria et al. (2023) used a convolutional neural network for increasing the spatial
resolution of simulated geostrophic ocean currents, helped by simulated sea surface temperature data. Liu and Wang (2021)
also used convolutional neural networks, this time for increasing the spatial resolution of low-resolution bands onboard the VI-
IRS (Visible Infrared Imaging Radiometer Suite) sensor, in order to obtain high spatial resolution ocean colour products. Kim
et al. (2023) and Lambhate and Subramani (2020) increased the spatial resolution of sea surface temperature data using Gen-
erative Adversarial Networks, and Zou et al. (2023) used a transformer model also with sea surface temperature. Barthélémy
et al. (2022) used a super-resolution data assimilation approach, based on an enhanced deep super-resolution network, to ingest
high spatial resolution observations into a hydrodynamical model. Peach et al. (2023) compared process-based and data-driven
approaches based on neural networks to increase the spatial resolution of wave forecasts, and Buongiorno Nardelli et al. (2022)
used a deep convolutional neural network to infer high spatial resolution ocean dynamics from satellite data. Applications are
very diverse in terms of used methodologies and variables. In this work, we are proposing a data-driven approach based on
DINEOF (Data Interpolating Empirical Orthogonal Functions, Beckers and Rixen (2003); Alvera-Azcárate et al. (2005)) to
increase the spatial resolution of Sentinel-3 ocean colour data using Sentinel-2 data. The aim is to obtain a unique dataset with
the temporal resolution of Sentinel-3 and the spatial resolution from Sentinel-2, by combining both data streams. DINEOF was
developed to interpolate missing data due to *e.g.* the presence of clouds, but as it will be shown here it can also, at the same
time, increase the resolution of the final, cloud-free dataset. DINEOF uses a truncated EOF basis to infer missing information
in satellite datasets hindered by the presence of clouds. The EOF basis extracts the dominant spatio-temporal variability and is
therefore an efficient approach to extract high spatial variability. As a data-driven technique, it is entirely based on the available
data, and does not need any a priori information about scales of variability or signal-to-noise ratio, or other input variables,
which makes its use easy and adaptable to any geophysical variable.

A huge challenge when working with several datasets to obtain a unique estimate of an ocean variable is the heterogene-
ity of the different sources: differences in the spectral bands present in each satellite (Blondeau-Patissier et al., 2014; Groom
et al., 2019), different spatial resolutions, and also a difference in the measurement time. This last factor can result in large
differences in dynamic regions, as is the case of the North Sea. In these regions, variables like chlorophyll-a concentration or
turbidity can experience large changes within a few hours due to the influence of strong tidal currents, storms and the wave
field (Fettweis et al., 2010; Wilson and Heath, 2019; Desmit et al., 2024), an influence that is, in addition, dependent on the
bathymetry. The region of study, the Belgian coast part of the North Sea (figure 1), is a shallow region characterized by a series
of sandbanks and dredging channels that influence water dynamics and bottom sediment resuspension. In this work, we aim at
minimising the differences due to the spectral characteristics of the Sentinel-2 and Sentinel-3, as it will be explained in section
2. The difference in time does not pose a problem in this study, since for a given day only one data source is used (Sentinel-2
if present or Sentinel-3 otherwise), and no merging of the two satellite datasets is performed.



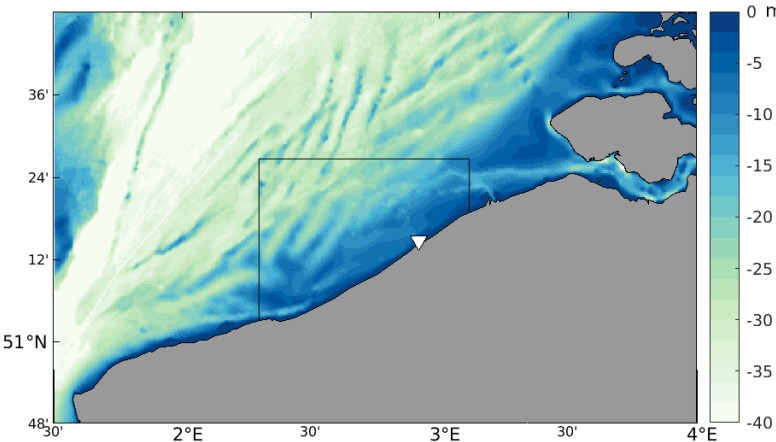

**Figure 1.** Bathymetry (m) of the southern North Sea. The black square shows the region used in this study and the white triangle shows the position of the validation station, RT1.

This work is organized as follows: section 2 describes the region of study, the satellite data and the in-situ data used. This section is followed by a description of the methodology to produce super-resolution data using DINEOF (section 3). The results, including the validation and an assessment of the scales resolved by all data sources, are presented in section 4. A description of small-scale variability in the southern North Sea using the super-resolution dataset is presented in section 5, and we conclude this work in section 6.

## 2  Data used

### 2.1  Study Area

This study focuses on dynamics and optically complex waters in the Belgian Coastal Zone (BCZ). The BCZ is a relatively shallow (<50 m) well-mixed shelf sea, connected to the North Sea in the north and to the English Channel in the west (Ruddick and Lacroix, 2006). It is characterized by a relatively high suspended sediment concentration, with a gradient from several hundreds of $g/m^3$ nearshore to $< 1g/m^3$ in the offshore waters, inversely related to the bathymetry (Nechad et al., 2009, 2011; Neukermans et al., 2012). Strong tidal currents and the tidal resuspension of sediments is the main cause of the high turbidity in the nearshore area (Fettweis and Van den Eynde, 2003; Fettweis et al., 2007). Annually recurring phytoplankton blooms are observed in spring and summer. These blooms are generally composed of diatoms and Phaeocystis globose (Lacroix et al., 2007). In recent years, blooms have been occurring earlier, likely in response to sea surface temperature increases and changes in nutrient outputs (Desmit et al., 2020; Alvera-Azcárate et al., 2021b). The water type is turbid coastal to turbid coastal with



high organic content. The water at the Research Tower 1 (RT1) near Oostende (51.24643°N, 2.91933°E), used in the validation
of the various datasets in this work, is turbid with tidal variability and with an occasional outflow from the port of Oostende
(Belgium) reaching the site.

## 2.2   Satellite data

The ocean colour satellite products used in this study are generated following the methodology applied in the Copernicus Ma-
rine high-resolution service using a multi-algorithm approach which aims at combining the best suited algorithm for different
water types. To facilitate Sentinel-2 and Sentinel-3 product generation, the processor is fully automated and set up in the DIAS
cloud environment CREODIAS (https://creodias.eu/). This processor starts from L1C and L1 data for Sentinel-2/MSI and
Sentinel-3/OLCI respectively and combines atmospheric correction processing (C2RCC + ACOLITE Dark Spectrum Fitting),
IDEPIX pixel classification, ocean colour algorithm application and product quality control (e.g. glint flagging, bottom reflec-
tion flagging, etc.) to provide Analysis Ready Data Layers (ARDL), i.e. chlorophyll (CHL), turbidity (TUR) and suspended
particulate matter (SPM). A schematic overview of the different processing steps taken is provided in figure 2.

### 2.2.1   Remote sensing reflectance and pixel classification

In order to obtain high-quality remote sensing reflectance (RRS) spectra for a large number of pixels while maintaining the
ability to handle both atypical water conditions and challenging atmospheric conditions, the atmospheric correction algorithms
ACOLITE-DSF (https://github.com/acolite/acolite) and C2RCC (https://c2rcc.org/) are combined to process L1C products to
L2R (Level 2 RRS products). While both algorithms have their strengths and weaknesses, they each use different approaches to
estimate RRS. C2RCC uses an underlying water reflectance model to fit the estimated RRS spectrum to a known form within
the boundaries of the training dataset. This method reduces noise in low RRS situations and provides greater retrieval power in
difficult circumstances, such as sun glinted and highly absorbing waters. On the other hand, ACOLITE-DSF does not assume
a specific water reflectance model, allowing it to return unusual Rrs spectra that correspond to optical properties not found
in typical water reflectance models. This can complement C2RCC where it is less performant, such as dredging plumes and
unusual algae blooms.


To combine the two approaches, a comprehensive, region-independent, and pixel-based automatic switching scheme is re-
quired, along with a technique for achieving a seamless transition between the two algorithms. The C2RCC to ACOLITE/DSF
pixel-based switching is described in detail in Van der Zande et al. (2023). Compatibility between the Sentinel-2/MSI products
and the Sentinel-3/OLCI products was ensured by applying the identical processing chain to both datasets.


Subsequently, the IDEPIX software (v2.2.10, algorithm update 8.0.3), available as a SNAP processor, is used for pixel clas-
sification, including cloud masking, cloud shadow identification, sea ice, floating vegetation, sub-pixel objects (ships, small



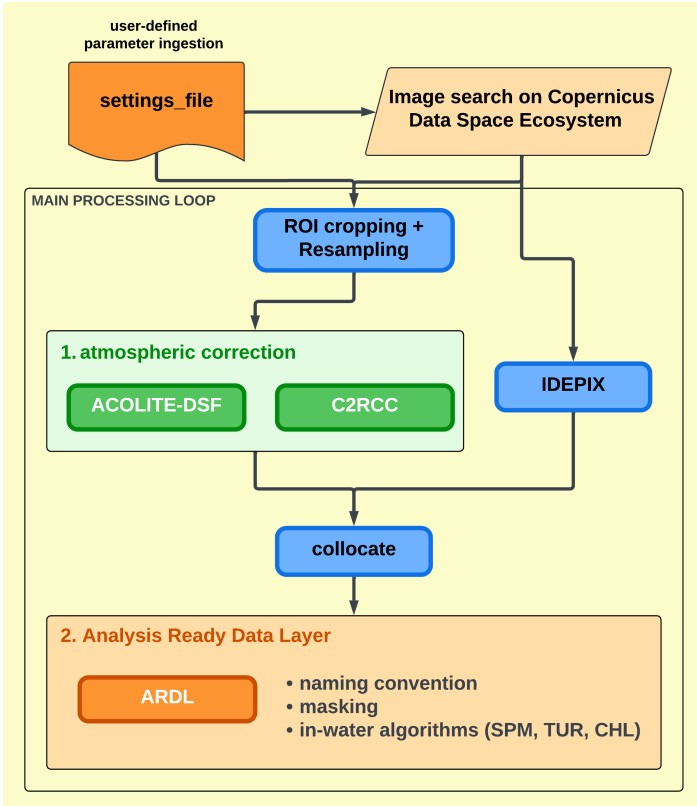

**Figure 2.** Processing steps applied for deriving L2 ARDL products, used for both the Sentinel-2/MSI and Sentinel-3/OLCI sensors. The processor combines the atmospheric correct algorithms ACOLITE and C2RCC, uses SNAP to crop and resample the data to the Region of Interest, runs IDEPIX for pixel classification after which all the intermediate layers are collocated together to the required resolution. In the final step, the in water ocean colour products are generated using specialized algorithms.

islands and rocks), and the land-water distinction taking temporary water bodies (e.g. intertidal areas, lagoons) into account. The IDEPIX processing step is complemented by quality tests coming from the atmospheric correction algorithms and additional tests (e.g. glint test).

### 2.2.2 Turbidity and Suspended Particulate Matter

The SPM and the TUR products were generated using the algorithm described by Nechad et al. (2010). While a single band can be used for TUR/SPM estimation, the optimal band depends on SPM concentration or TUR levels – if RRS is too low (e.g. for longer wavelengths in low SPM waters) then TUR/SPM estimation will be significantly affected by noise or errors in RRS – If RRS is too high (e.g. for shorter wavelengths in high TUR/SPM waters) then the saturation phenomenon means



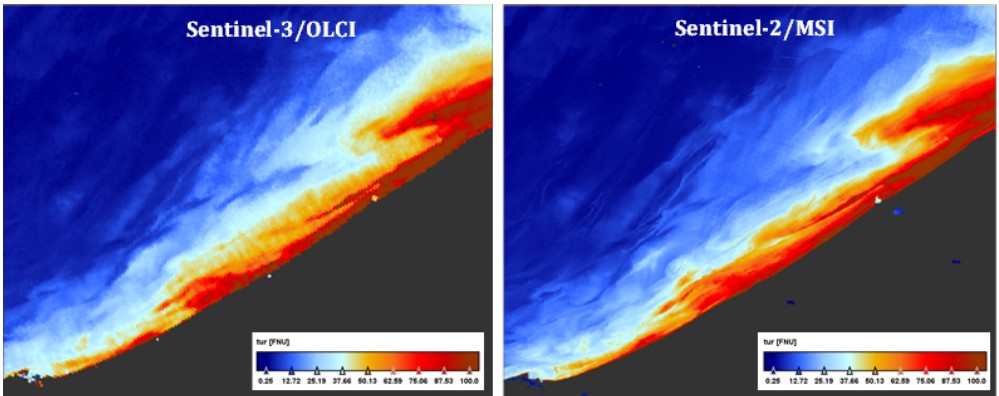

**Figure 3.** TUR products for 5 April 2020 for the Belgian Coastal Zone region generated using the multi-sensor processor using Sentinel-3/OLCI (left) and Sentinel-2/MSI (right) source data.

that RRS becomes insensitive to changes in TUR/SPM. This has led to the development of "switching single band algorithms" (Novoa et al., 2017) using the basic single band formulation of Nechad et al. (2010) but with different wavelengths used at different SPM concentrations and typically a smooth weighting between two adjacent spectral bands to avoid image artefacts. The Novoa et al. (2017) approach is applied to both the SPM and TUR products providing a multi-band SPM and TUR product using two bands (red: 665 nm and near-infrared: 865 nm). An example of the TUR products for the Belgian Coastal Zone region is provided in figure 3 showing a good correspondence between both the Sentinel-2/MSI and Sentinel-3/OLCI products providing information at different spatial and temporal resolutions.

### 2.3 Multi-sensor chlorophyll data

For the scale assessment described in section 4.3, daily chlorophyll data at a spatial resolution of 1 km is used. These data are obtained from the level-3 multi-sensor cmems_obs-oc_atl_bgc-plankton_my_l3-multi-1km_P1D CMEMS product. This product includes data from different sensors (SeaWIFS, MERIS, MODIS-Aqua, MODIS-Terra, VIIRS-SNPP, VIIRS-JPPS1, OLCI-S3A and OLCI-S3B) and covers the whole North Sea area (48.46°N - 55.96°N; -1.64°E - 6.15°E) from 1 February 2022 to 1 November 2022. A total of 271 images are available, with an average amount of missing data, due to cloud cover and quality control, of 38.17%.

### 2.4 In situ data

The RRS products were validated using Pan-and-Tilt Hyperspectral Radiometer system (PANTHYR, Vanhellemont (2020); Vansteenwegen et al. (2019)) for the period 2019-2022. An autonomous PANTHYR system has been deployed at the Research Tower 1 (RT1) near Oostende (RT1, 51.24643°N, 2.91933°E). The PANTHYR system has two TriOS RAMSES radiometers mounted on a pan-and-tilt head, one for up- and downwelling spectral radiances, and one with a cosine collector to measure





spectral irradiance, enabling to determine the RRS signal. The PANTHYR measures autonomously every 20 min at programmed relative azimuth angles to the sun. In the present study, measurements were made at a 270° azimuth angle. Because of the hyperspectral measurements, one significant advantage of the PANTHYR datasets when compared to AERONET-OC,

typically used for validating ocean-colour satellites (Zibordi et al., 2009), is that the hyperspectral instrument permits the validation of all MSI and OLCI VNIR bands within the 400-900 nm range, including several nearinfrared (NIR) bands not available with the AERONET-OC instrument. RRS data were finally convolved to the relative spectral response functions of the MSI and OLCI instruments on Sentinel-3 A and B and Sentinel-2 A and B.

Matchups for PANTHYR stations were extracted from shifted locations near the deployment tower, to avoid platform effects such as direct pixel contamination and shadows, as well as in-water wakes. For the match-up extraction, a maximum time difference of 2 hours between in situ observation and satellite overpass was allowed. However, the high frequency measurement protocol from the in-situ measurement stations resulted in shorter time differences between in situ and satellite observations. The matchup validation protocol described by Bailey and Werdell (2006) was applied to remove erroneous matchups from the

analysis. Macro-pixels of 3x3 60m pixels for Sentinel-2/MSI and 3x3 300m for Sentinel-3/OLCI were extracted from the L2 products.

## 3  Methodology

### 3.1  DINEOF

DINEOF (Data Interpolating Empirical Orthogonal Functions, Beckers and Rixen (2003); Alvera-Azcárate et al. (2005)) is used to calculate the reconstruction of the data and to enhance the resolution of the combined Sentinel-2 and Sentinel-3 dataset. DINEOF computes the missing data information from a three-dimensional dataset by calculating a truncated Empirical Orthogonal Function (EOF) basis. These EOFs are calculated increasingly (starting from one until the optimal number of EOFs is found) and iteratively until a convergence is reached, and at each iteration the estimate of the missing data is updated

using the latest EOF modes. About 3% of valid data (in the form of clouds, Beckers et al. (2006)) is masked at the beginning of the procedure, and these data are used to determine the number of EOFs that minimizes the reconstruction error (in terms of its Root Mean Square Error, RMSE). When three consecutive EOFs provide an increasingly higher RMSE, the procedure is stopped and the final reconstruction is performed with the optimal amount of EOFs determined by this cross-validation.

In addition to the reconstruction of missing data in several variables (*e.g.* sea surface temperature (Alvera-Azcárate et al., 2005), chlorophyll (Alvera-Azcárate et al., 2021b), turbidity (Alvera-Azcárate et al., 2015), salinity (Alvera-Azcárate et al., 2016) as well as multivariate reconstructions (Alvera-Azcárate et al., 2007)), DINEOF has been used to detect outliers in satellite data (Alvera-Azcárate et al., 2012, 2015) and shadows in high spatial resolution satellite data (Alvera-Azcárate et al., 2021a). DINEOF has therefore shown it can perform a wide range of analysis of satellite data with the aim of improving their





quality and completeness.

## 3.2   Generation of super-resolution data

In order to obtain a super-resolution reconstruction from the combination of Sentinel-3/OLCI and Sentinel-2/MSI data mentioned in section 2, the initial gappy matrix is prepared as follows: Sentinel-2 data are used on days when they are available,

and on days with no Sentinel-2 coverage, Sentinel-3 data, linearly interpolated at the Sentinel-2 spatial resolution, are used. The matrix consists therefore of a mixture of Sentinel-2 and Sentinel-3 data. The interpolation of Sentinel-3 data onto the Sentinel-2 grid is done to preserve the size of the matrix, which has to be constant in order to be used in DINEOF, and also to determine the spatial resolution of the final dataset, but no gain in resolution is done at this step.

On days when both Sentinel-2 and Sentinel-3 data are available, only Sentinel-2 data are used. We could slightly increase the spatial coverage on these days by combining both datastreams but we had found the increase to be very small, since cloud distribution does not change substantially between the Sentinel-3 and Sentinel-2 passes. In addition, depending on the tidal currents, the difference in turbidity between the two satellite passes can be quite high, and this can lead to discontinuities between both estimates, which in turn can be reflected in the final dataset. We have decided therefore to avoid this problem by

using a unique data source on a given day.

  The initial dataset with the combined Sentinel-2 and Sentinel-3 data has 210 time steps, of which 63% are Sentinel-3 data and 37% are Sentinel-2 data. Days with too high cloud coverage (more than 98% of missing data over the study region) are not used, which brings the final size to 163 days, distributed from 18 January to 17 December 2020.


  The combined dataset is fed to DINEOF as a unique matrix. As DINEOF performs all calculations on the initial grid, it is important that the initial data are already interpolated to the final grid that we want to obtain (i.e. the high spatial resolution grid). Through the EOF basis calculation, the high spatial information of Sentinel-2 is extracted by the EOFs, and this information is then projected into the final dataset by the EOF basis, effectively increasing the spatial resolution of the initial dataset.


## 4   Results

### 4.1   Super-resolution data

Super-resolution DINEOF (section 3) has been applied to the turbidity data in the Oostende region described in section 2. The initial matrix has 45% of missing data. DINEOF has been applied to the combined Sentinel-2 and Sentinel-3 data for 2020 and

23 EOFs were retained as optimal for the reconstruction, with a cross-validation error of 1.4 FNU. The size of the matrix was





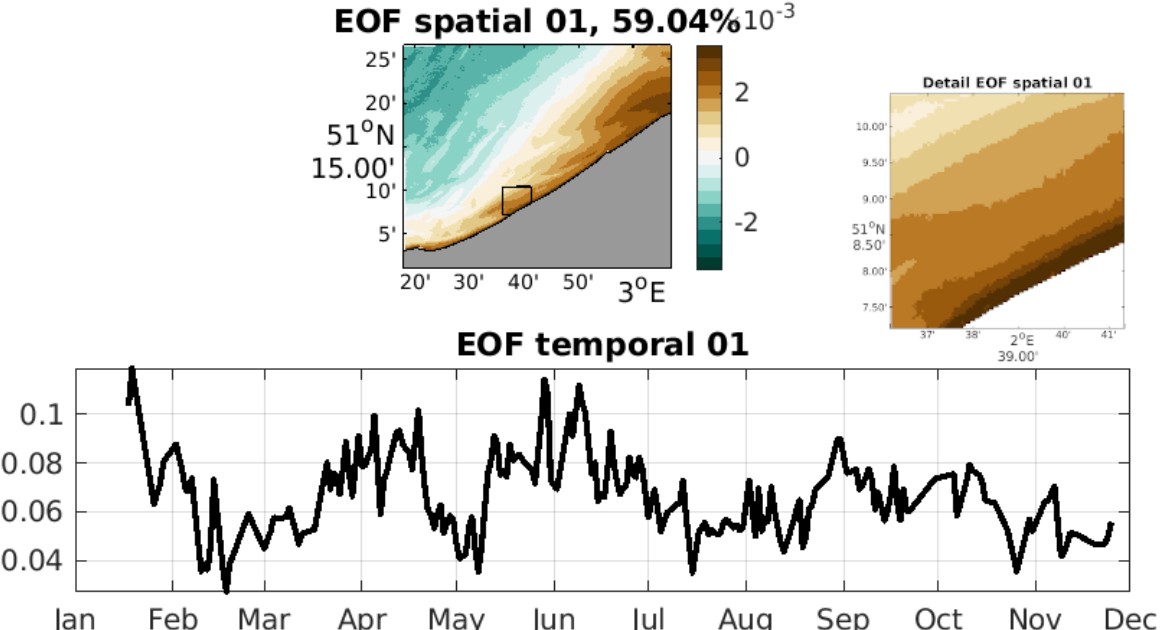

**Figure 4.** First EOF mode of the 2020 turbidity data obtained by DINEOF. Top left: spatial EOF mode. Top right: detail of the spatial mode on the black spare shown in the left panel. Bottom: temporal EOF mode.

$946 \times 789 \times 210$ (longitude, latitude and time), and the DINEOF run took 8 hours to complete (on an Intel Core i9-10900X CPU at 3.70GHz). The only two optional parameters to set in DINEOF are related to the filtering of the temporal covariance matrix (Alvera-Azcárate et al., 2009) and were fixed to $\alpha = 0.01$ (the amplitude of the filter) and $n = 3$ (the number of iterations of the filter). These result in a filter of about 1.1 days. The first three EOFs explain, respectively, 59.04%, 25.73% and 3.74% of

the total variability of the dataset, and the 23 EOFs retained explain a total of 97.9% of the variability. When analysing the spatial and temporal structure of these EOFs, we can observe the influence of small scale variability contained in them. The first EOF mode (figure 4) displays an inshore-offshore gradient, in which more turbid waters are found in the nearshore region. An increasing turbidity along the coast is found towards the region of the Scheldt river estuary. The factors responsible for higher turbidity in this region have been attributed to tides and meteorological conditions (Fettweis and Van den Eynde, 2003),

and show peaks in April, June and September-October 2020. This mode shows large-scale processes, and there is no apaprent small-scale variability present in the first spatial EOF.

The second EOF mode (not shown) already displays small-scale spatial variability. We show here the third EOF mode (figure 5), which shows a northeast-southwest gradient, with high variability in the temporal mode. On the detail panel shown

in the figure we can also appreciate a thin region along the coast with oposite behaviour than the more offshore waters. It has been shown (https://www.esa.int/Applications/Observing_the_Earth/Copernicus/Sentinel-2/Near-shore_phytoplankton_





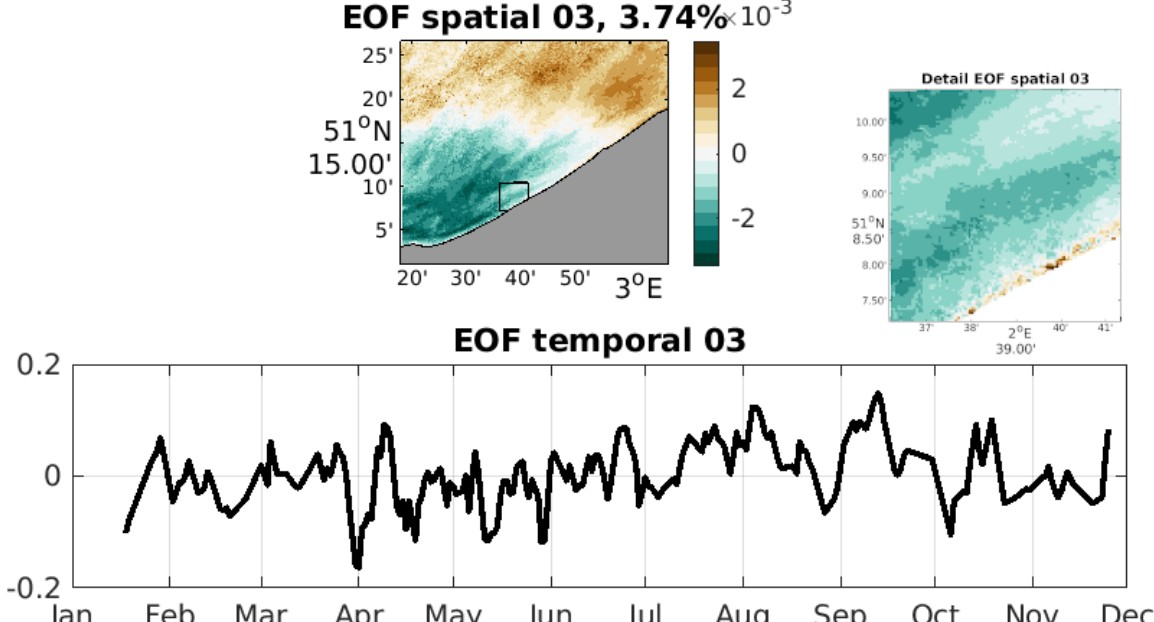

**Figure 5.** Third EOF mode of the 2020 turbidity data obtained by DINEOF. Top left: spatial EOF mode. Top right: detail of the spatial mode on the black spare shown in the left panel. Bottom: temporal EOF mode.

bloom_captured_from_space) that Sentinel-2 is able to capture small scale varibility that was previously unknown, thanks to its high spatial resolution, a variability that has been captured by the EOF basis to produce the final, super-resolution datasets that will be shown in this section. The validation of the initial data and the super-resolution reconstruction will be presented in
section 4.2.

Figure 6 shows the initial turbidity data on 9 May 2020, a day with initially Sentinel-3 data (hence, low resolution data). A day with low cloud coverage has been chosen in order to show the capability of DINEOF to enhance the spatial resolution. The initial data (figure 6 top left panel) show a series of high and low turbidity regions, a pattern that is due to the presence of
sandbanks close to the Belgian coast. The changes in depth in this region induce large differences in turbidity from day to day. High turbidity values in the east of the figure are due to the Scheldt-Rhine river plume, located to the east of the domain. The reconstructed image (figure 6 top right panel) is able to retain most of this variability. A north-south transect is also included in the figure to show the small scale variability that has been included in the reconstruction. The initial data (in blue in the bottom panel of figure 6) does not contain this small scale variability, and presents instead a step-like nature due to the low spatial
resolution of the initial dataset. This step-like variability is absent from the final, super-resolution data, which instead shows smaller-scale variability.





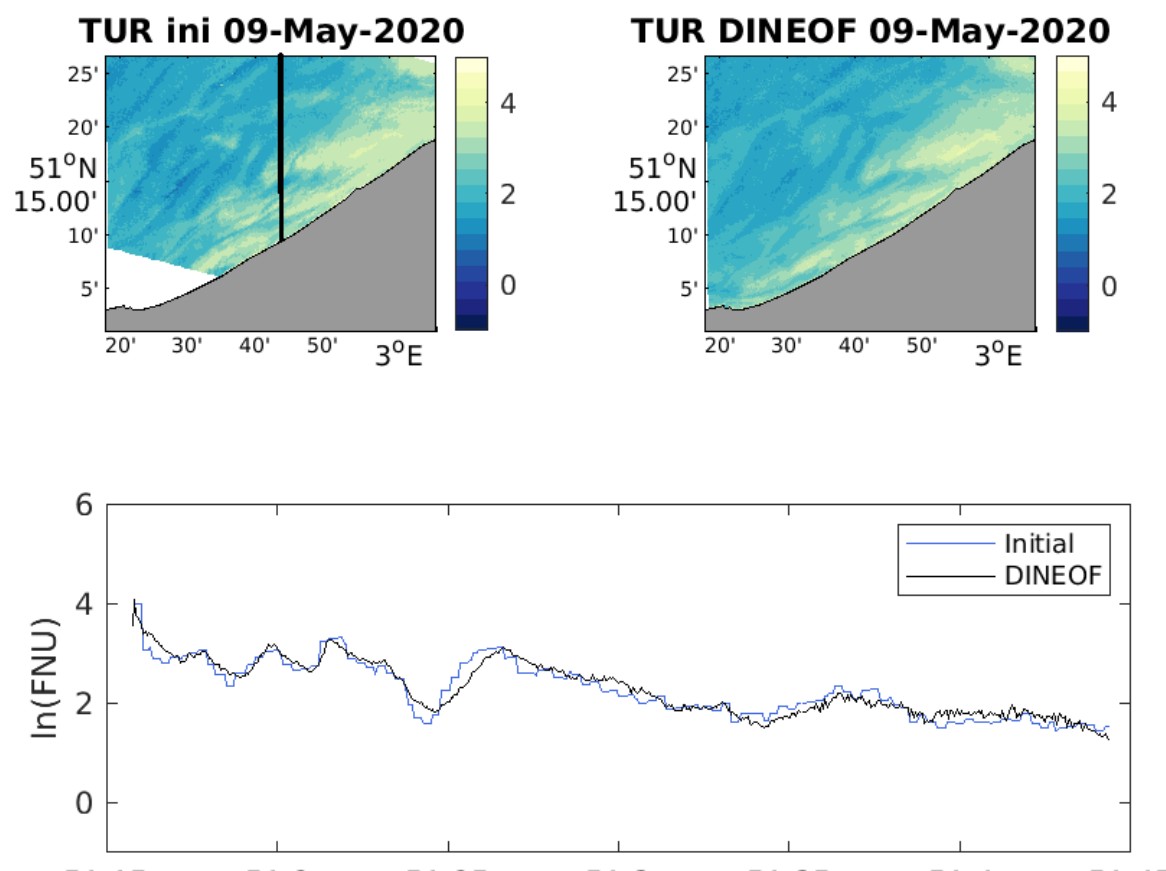

**Figure 6.** Top left: initially cloudy data at 300 m resolution, on 9 May 2020. Top right: DINEOF run of the mixed Sentinel-2 and Sentinel-3 dataset, at 100 m resolution. Bottom: north-south transect for the two datasets (blue: initial data at 300 m; black: super-resolution DINEOF reconstruction). All plots are in logarithmic scale.

Figure 7 shows a similar reconstruction as in figure 6, but this time for a date in which Sentinel-2 data are available (hence, high spatial resolution data). This example shows that the variability of the super-resolution DINEOF reconstruction is similar to the one of the initial dataset, and there is only a limited amount of variability lost with the analysis, mostly in regions with low values of turbidity (figure 7 bottom panel). The presence of high turbidity regions caused by the presence of sandbanks is also visible in this image, and a strong inshore-offshore turbidity gradient can be seen, which is well retained by the reconstruction. Turbid waters in the eastermost part of the domain, close the Scheldt estuary and the shallowest region of the domain, can





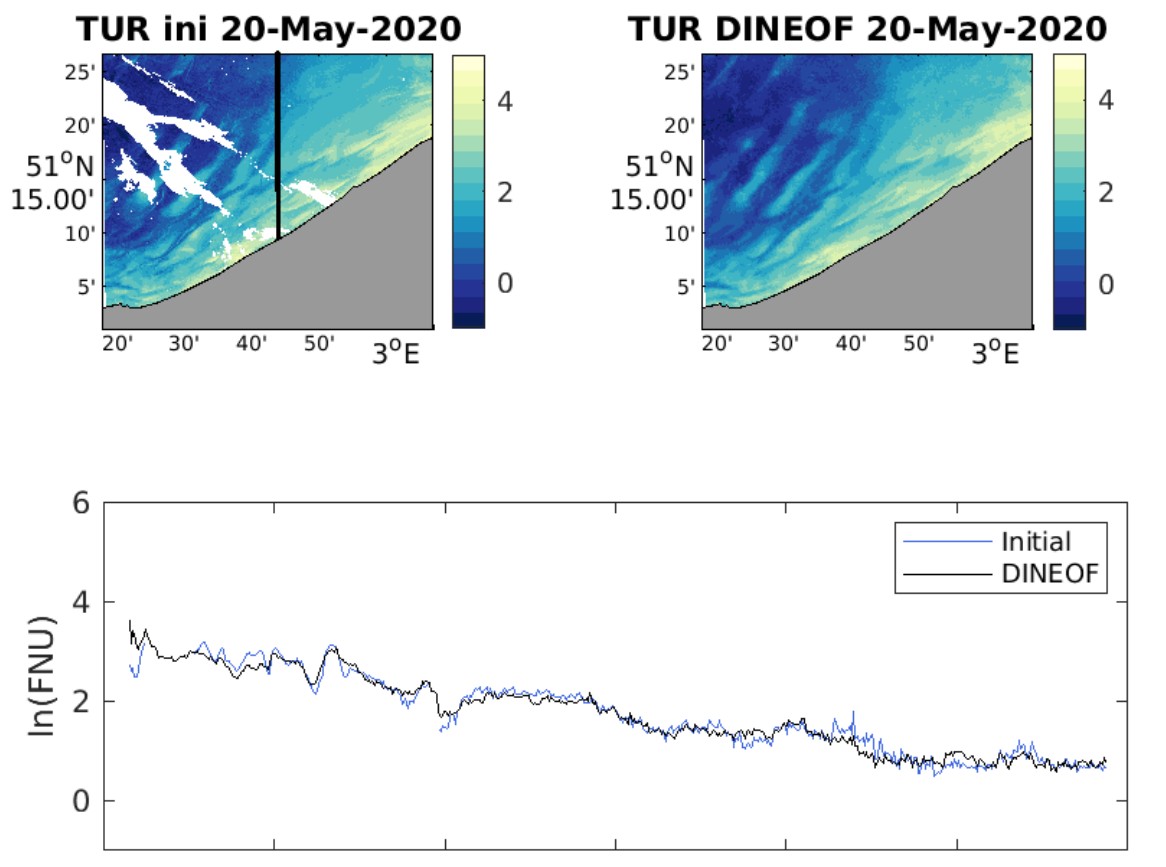

**Figure 7.** Top left: initially cloudy data at 100 m resolution, on 20 May 2020. Top right: DINEOF run of the mixed Sentinel-2 and Sentinel-3 dataset, at 100 m resolution. Bottom: north-south transect for the two datasets (blue: initial data at 100 m; black: super-resolution DINEOF reconstruction). All plots are in logarithmic scale.

also be seen.


The results shown so far are from images that have good data coverage, but DINEOF also provides super-resolution data on days with high cloud coverage. An example is shown in figure 8, corresponding to 12 April 2020. The initial data is missing for most of the domain, but DINEOF is still able to provide a reconstruction with a good spatial variability, as shown both in the spatial map with the presence of high turbidity associated with the river plume and the sandbanks (top right panel of figure

8) and in the north-south transect in the bottom panel. The reconstruction of the turbidity spatial distribution on days with high



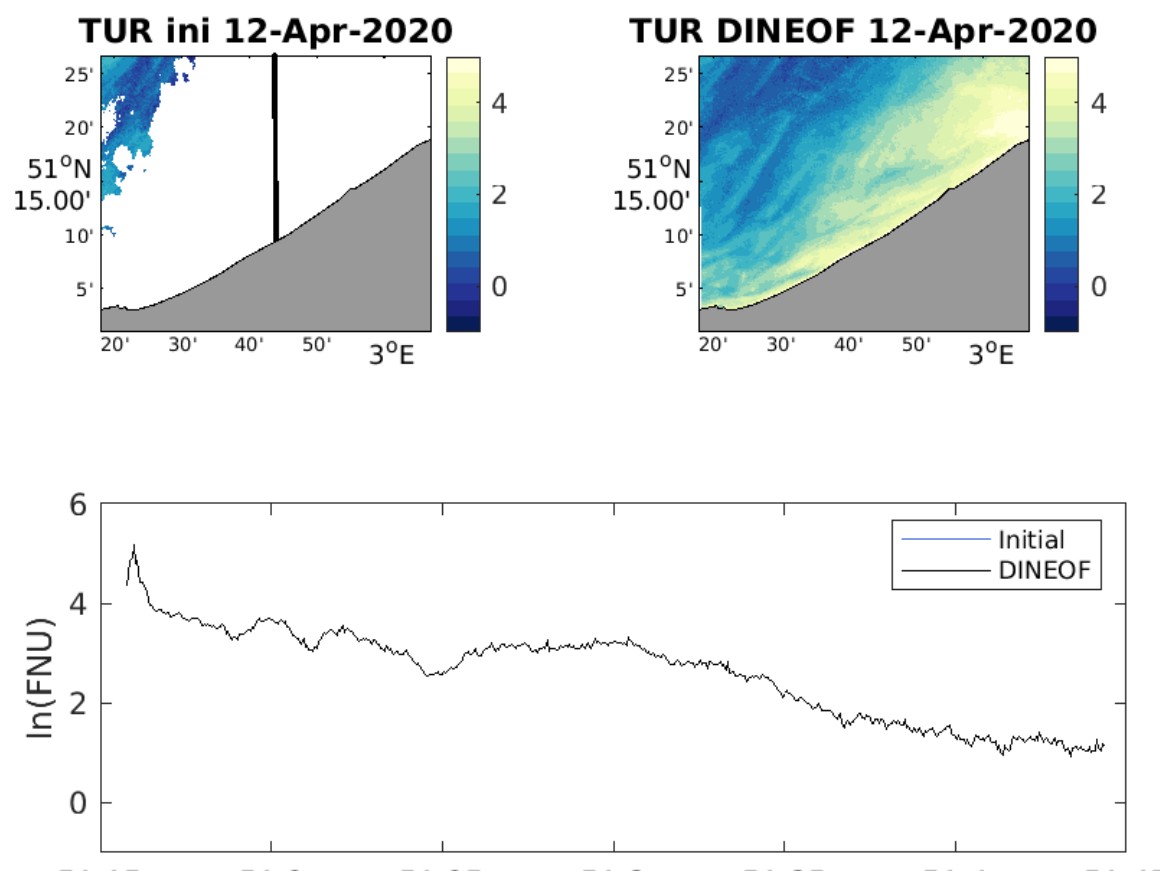

**Figure 8.** Top left: initially cloudy data at 100 m resolution, on 12 April 2020. Top right: DINEOF run of the mixed Sentinel-2 and Sentinel-3 dataset, at 100 m resolution. Bottom: north-south transect for the two datasets (blue: initial data at 100 m; black: super-resolution DINEOF reconstruction). All plots are in logarithmic scale.

amounts of missing data is possible because of the three-dimensional nature of DINEOF, which exploits the spatio-temporal coherence of the data and enhances temporal correlations (Alvera-Azcárate et al., 2009).

At some moments, there can be outliers or noise in the initial dataset, despite the strong quality controls applied to the data. The fact that DINEOF uses a truncated EOF basis to compute the missing data results in a partial loss of variability. However, this truncated EOF basis guarantees that the presence of outliers does not influence the overall quality of the reconstruction. As an example, in figure 9 we can see the turbidity on 28 May 2020, with a region in the northern part affected by the presence



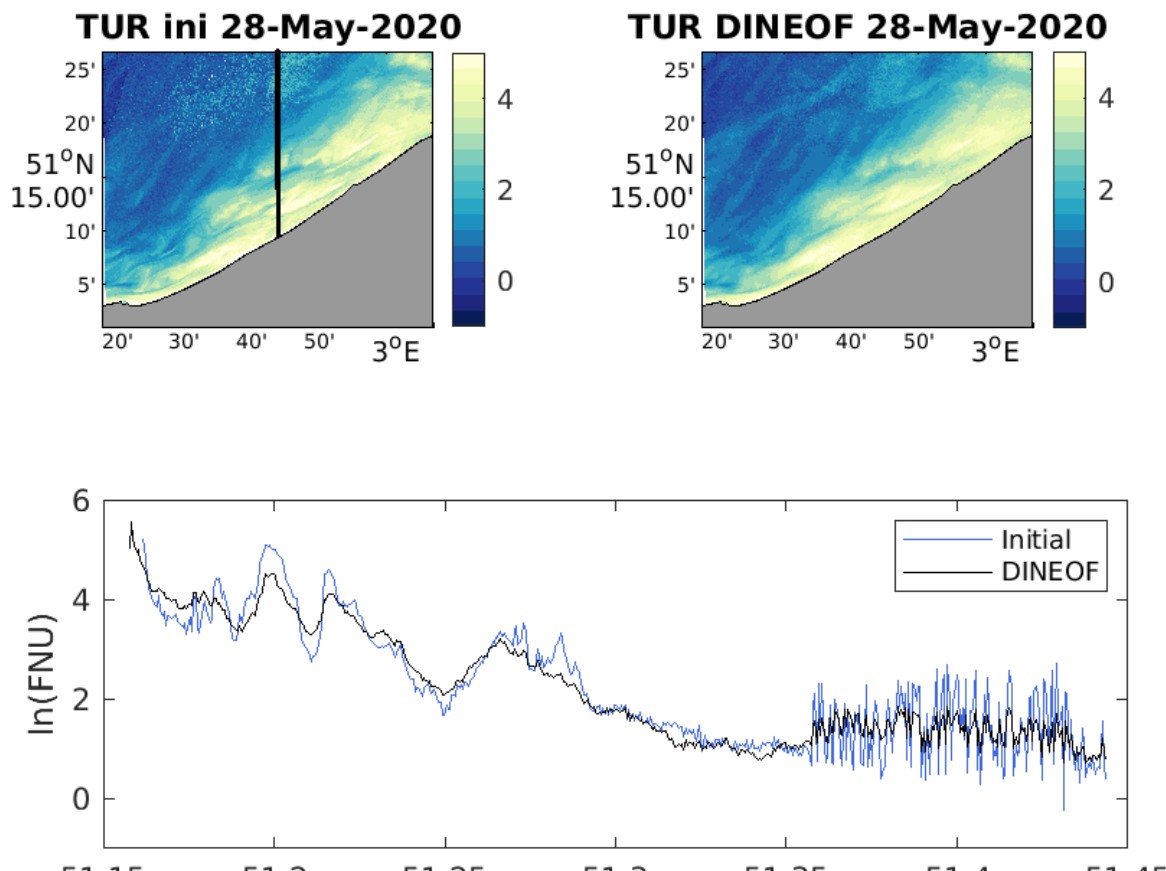

**Figure 9.** Top left: initially cloudy data at 100 m resolution, on 28 May 2020, with a noisy region in the northern part of the domain. Top right: DINEOF run of the mixed Sentinel-2 and Sentinel-3 dataset, at 100 m resolution. Bottom: north-south transect for the two datasets (blue: initial data at 100 m; black: super-resolution DINEOF reconstruction). All plots are in logarithmic scale.

of noisy data. These are probably due to non-detected thin clouds. The north-south transect shows that the variability of these data far exceeds normal variability expected for the region. The DINEOF super-resolution results provide a reduced amount of noise and an improved quality of the final product, while still providing an accurate depiction of small scale variability.



## 4.2 Validation

Using the in-situ data described in section 2.4, a quality assessment of the initial data and the DINEOF results has been realised. Figure 10 shows the RRS matchups between Sentinel-2/MSI and the PANTHYR in situ instrument installed on RT1. In the considered deployment period between 2019-12-11 and 2023-08-01, 528 MSI L1C images were available for processing with 59 matchups with the PANTHYR instrument which passed the quality flagging of the individual ACs (i.e. ACOLITE-DSF and C2RCC), IDEPIX quality flagging and the match-up quality flagging. When using the merged approach for atmospheric correction, the best performing bands are 492, 560, 665, and 704nm which are typically used for retrieval of turbidity and turbid water chlorophyll-a. For these bands the MAPE ranges between 9.88% and 17.20%.

Figure 11 shows the RRS matchups between the Sentinel-3/OLCI and the PANTHYR instrument installed on RT1. In the deployment period between 2019-12-11 and 2023-08-01, 2334 OLCI L1FR images were available for processing with 179 common matchups with the PANTHYR instrument which passed the quality flagging of the individual ACs (i.e. ACOLITE-DSF and C2RCC), IDEPIX quality flagging and the match-up quality flagging. For the merged approach of atmospheric correction, the best performing bands are 443, 492, 560, 665, and 709nm which are typically used for retrieval of turbidity and turbid water chlorophyll-a. For these bands the MAPE ranges between 10.74% and 19.00%.

Figure 12 provides a graphical overview of the band-specific statistical metrics (i.e. slope, MAPE and RMSE) for the PANTHYR matchup analysis for both the Sentinel-2/MSI and Sentinel-3/OLCI matchups illustrating that the identical processing chain applied to both satellite datasets results in coherent validaton results.

The accuracy of the DINEOF super-resolution products was validated for the Belgian Coastal Zone region by using the hyperspectral in situ data set from the autonomous PANTHYR systems deployed at Research Tower 1 (RT1) near Oostende to generate an in-situ turbidity product which was directly compared with the satellite derived turbidity products. The in-situ turbidity product was generated with the same algorithm as used for the satellite products. Figure 13 shows the turbidity time series for 2020 overlaying the in-situ data, both the Sentinel-2 and Sentinel-3 turbidity products and the final super-resolution DINEOF gap-filled product showing that the DINEOF product is able to capture the in situ turbidity signal very well between March and September. In January and February the DINEOF product shows slightly lower values which can be caused by the fact that in those months the availability of cloud-free satellite products from Sentinel-2 and Sentinel-3 is very scarce.

An objective intercomparison was achieved by a match-up analysis. For the match-up extraction, a maximum time difference of 1 hour between in situ observation and satellite overpass was allowed. The matchup validation protocol described by Bailey and Werdell (2006) was applied, to remove erroneous matchups from the analysis based on macro-pixels of 3x3 pixels from the satellite turbidity products. Figure 14 shows the results of the match-up analysis for Sentinel-2, Sentinel-3 and the gap-filled DINEOF super resolution products. These graphs show that both the Sentinel-2 and Sentinel-3 products show a





**Figure 10.** Validation of remote sensing reflectance (RRS) for Sentinel-2/MSI data. Hyper-spectral in situ stations from the HYPERNET network in Ostend (Belgium) were used.





Figure 11. Validation of remote sensing reflectance (RRS) for Sentinel-3/OLCI data. Hyper-spectral in situ stations from the HYPERNET network in Ostend (Belgium) were used.





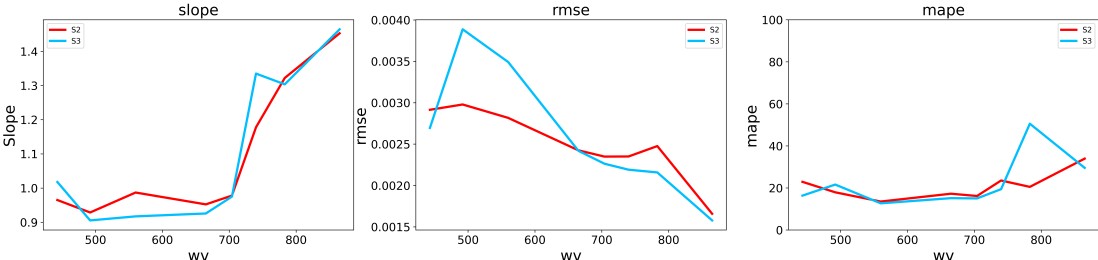

**Figure 12.** Overview of band-specific statistical metrics (slope, root mean square error -rmse-, mean average percentage error -mape-) for the RRS matchups comparing Sentinel-3/OLCI (S3) with Sentinel-2/MSI (S2)

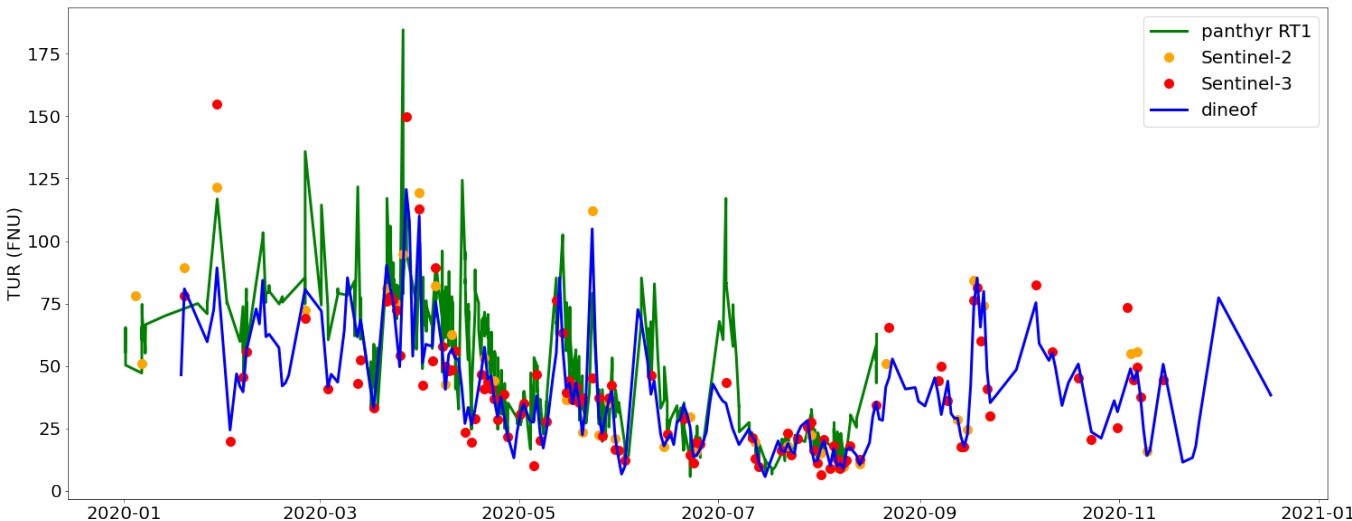

**Figure 13.** Turbidity time series for 2020 at the RT1 Hypernets station generated using the in situ Hyperspectral Panthyr data (green line), the Sentinel-2 satellite data (orange dots), the Sentinel-3 satellite data (red dots) and the gap-filled DINEOF super resolution satellite product (blue line).

good agreement with the in-situ observations with mean average percentage differences around 6%. The temporal frequency of Sentinel-3 overpasses over the region of interest results in more than 3 times more matchups. Considering the DINEOF super resolution matchups, this number of matchups is increased by another 34% with very similar statistics compared to the Sentinel-3 matchups, showing DINEOF's ability to retain the turbidity information provided by the source products. We do see a slight underestimation of the turbidity values by both satellites, especially for higher values (turbidity > 50 FNU).



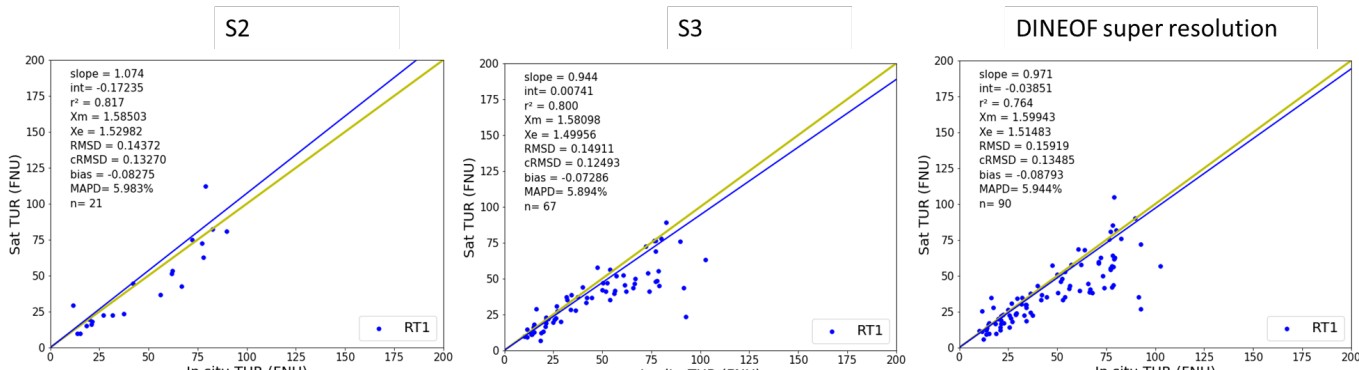

**Figure 14.** Matchup results of daily Sentinel-2, Sentinel-3 and DINEOF super resolution TUR products against in situ observations obtained from the autonomous PANTHYR system at the RT1 station in the Belgian Coastal Zone.

## 4.3 Scale Assessment

In order to determine which scales are reconstructed in the super-resolution DINEOF approach, a test using the multi-sensor satellite chlorophyll data at 1km spatial resolution described in section 2.3 was performed. These data are downscaled to 5 km using a near neighbourgh interpolation (to avoid smoothing the data). Following the same procedure as with the combined Sentinel-2 and Sentinel-3 turbidity dataset, we intercalate the 1 km spatial resolution data and the 5 km spatial resolution data. The ratio of this mixed dataset is 1 high resolution image (at 1 km) for every 3 low resolution images (at 5 km), to mimic the ratio of the Sentinel-2 and Sentinel-3 combination used in the previous section. This allows us to compare the scales reconstructed on this dataset with the initial 1 km resolution data, both on days in which 1 km data and 5 km data were used. In addition, we have also made a reference reconstruction of the original, 1 km spatial resolution data, so that we can assess how a full high spatial resolution reconstruction compares with the mixed dataset reconstruction.

The DINEOF reconstruction of the mixed dataset used 22 EOFs, and an example of reconstruction is shown in Figure 15. On this date, 9 July 2022, the initial dataset has a spatial resolution of 1 km. A north-south transect shows that the reconstruction (in blue in the figure) is able to reproduce the variability of the initial data (in black in the figure). The reconstruction of the original dataset (using only 1 km resolution data) is also shown for reference (in green in the figure). The variability observed in the north-south transect for the reference run and the super-resolution reconstruction are very similar, showing the capability of the super-resolution approach to retain small-scale variability.

On a day with initially 5 km spatial resolution data (Figure 16 on 28 July 2022), we can see that the initial data shows a step-like variability in the bottom panel of Figure 16. The reconstructed data (in blue) is able to increase the variability of the 5 km data to mimic the variability of the reference dataset (in green), effectively increasing the spatial resolution of the results. The spatial distribution of chlorophyll is similar in all figures shown, and it is difficult to determine any differences between



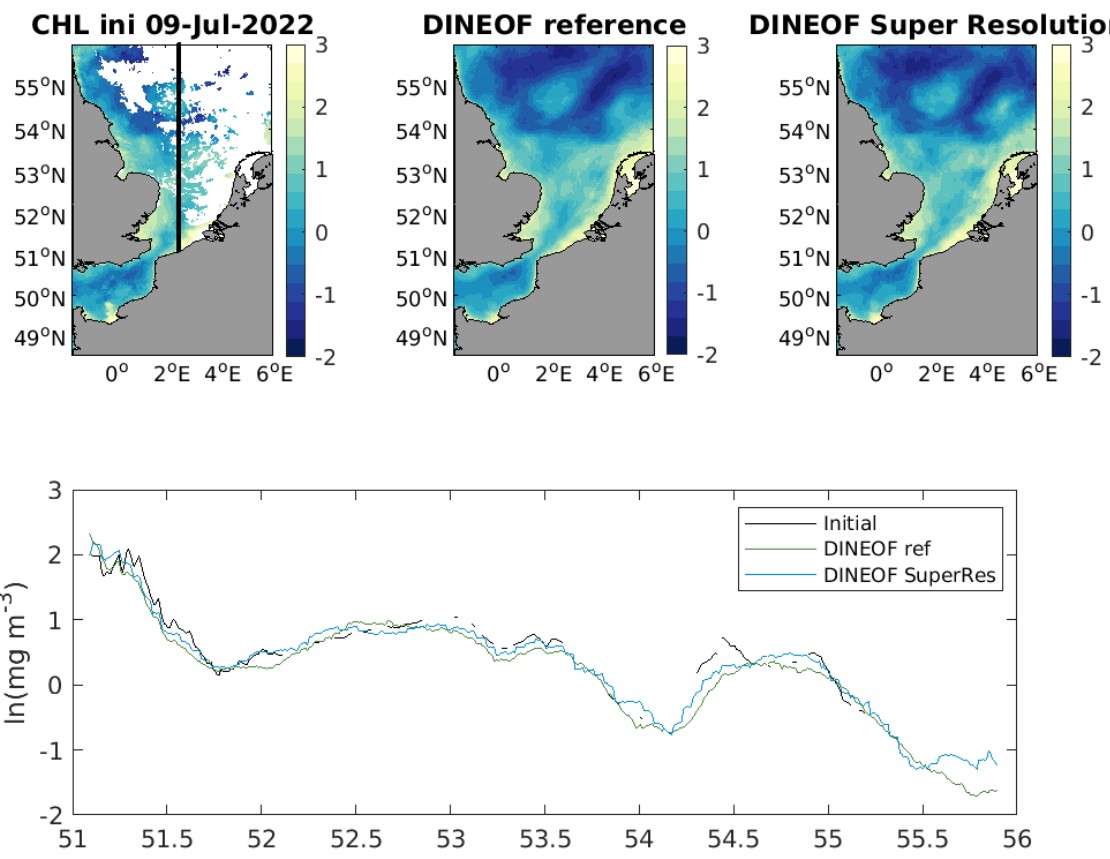

**Figure 15.** Top left: initially cloudy data with a 1 km resolution, on 9 July 2022. Top center: DINEOF reconstruction of the 1 km data (reference run). Top right: DINEOF run of the mixed dataset. Bottom: north-south transect for the three datasets (black: initial data at 1 and 5 km; green: reference run at 1km; blue: super-resolution run).

the different datasets over this large domain. In order to see the effect of the super-resolution DINEOF approach we need to
look at a small region. Figure 17 shows a detail of the reconstruction on 1 June 2022, in which initially 5 km data are present.
The super-resolution DINEOF reconstruction (in the right panel of Figure 17) is shown to provide higher spatial resolution
than the initial data, similar to what is obtained with the reference reconstruction at 1 km (shown in the central panel of Figure
17), despite the coarse resolution of the initial field. There are some remaining edge effects showing the initial 5 km grid in
the super-resolution, but this comes from the choice of the near-neighbourgh interpolation. A linear interpolation would avoid
such pattern in the final results.



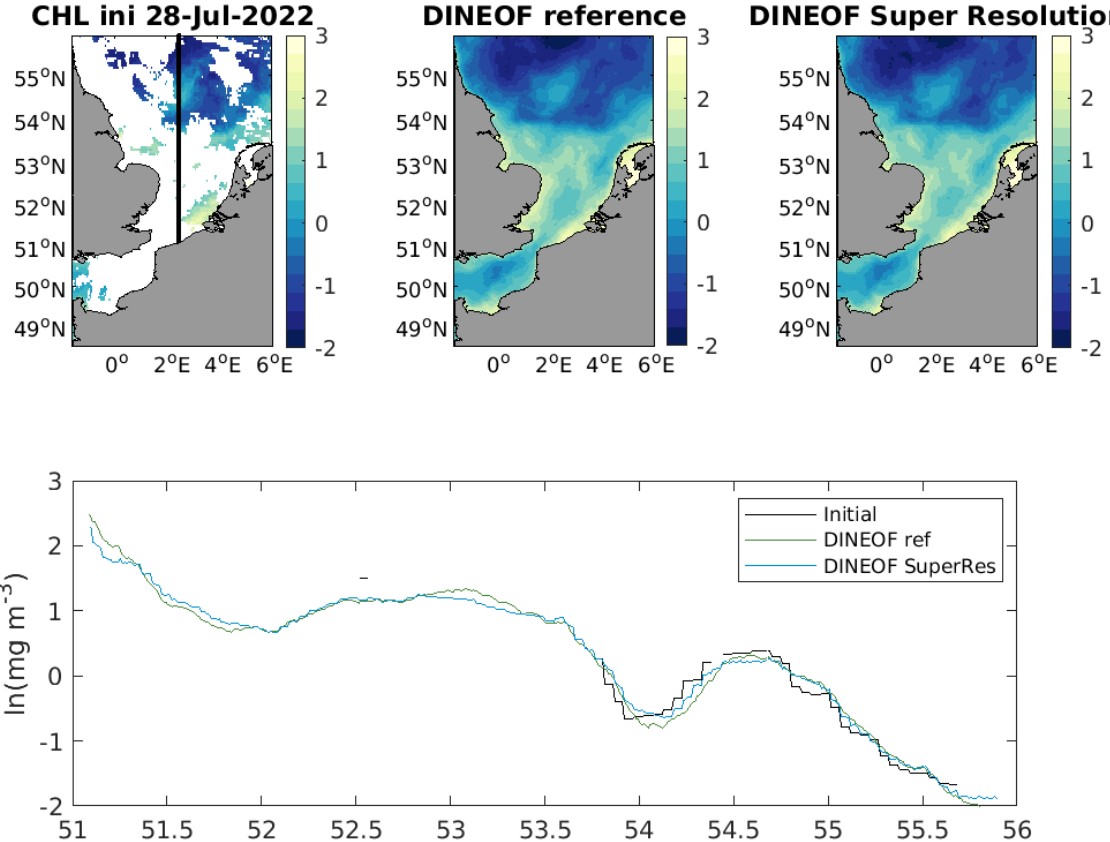

**Figure 16.** Top left: initially cloudy data with a 5 km resolution, on 28 July 2022. Top center: DINEOF reconstruction of the 1 km data (reference run). Top right: DINEOF run of the mixed dataset. Bottom: north-south transect for the three datasets (black: initial data at 1 and 5 km; green: reference run at 1km; blue: super-resolution run).

## 5   Submesoscale variability in the Belgian Coastal Zone

The super-resolution data obtained in this work allow us to analyse the variability of turbidity at the submesoscale in the study region. The river sediments carried out by the Scheldt and the resuspension of bottom sediments at the along-shore sandbanks are the two major contributors to small-scale variability of turbidity in this region.

Sandbank-induced high turbidity patterns in the Belgian coast are influenced by the topography, and horizontal water movement due to tidal currents results in rapid particle deposition outside of these shallow environments. As a result, turbidity is often high inside the sandbank region and up to about 10 m depth as observed for example in figure 18. We use hourly surface



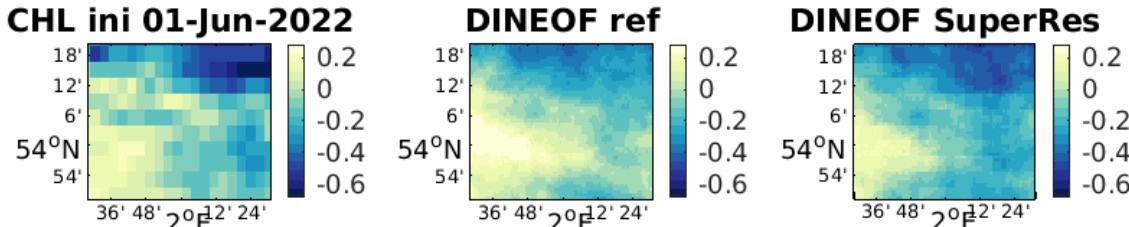

**Figure 17.** Example of super-resolution DINEOF reconstruction. Left panel: initial data downgraded to 5 km spatial resolution on 1 June 2022. Center panel: reconstruction of the reference run at 1 km spatial resolution. Right panel: reconstruction of the 5 km data with DINEOF using the mixed 1 km and 5 km dataset.

currents obtained from Legrand and Baetens (2021) to assess their influence on turbidity distribution. The intensity of currents and their direction in the hours preceding the time of the satellite pass have a large influence on the average turbidity values over the region, like on 16 October (figure 19) which presents a similar tidal phase than 19 May (top right image) but with stronger currents, resulting on an overall higher turbidity over the whole region.

The bottom panels in figures 18 and 19 show that there is an overall decreasing turbidity in the offshore direction, with similar variability at all depths. The effect of the presence of sandbanks in the resuspension of turbidity is also visible in these images, with regions of higher turbidity corresponding to the presence of these sandbanks. During weak water currents periods (Figure 18), the effect of the sandbanks on water turbidity is clearly seen, with sediments depositing at the deeper, inter-sandbank regions. During strong water current periods (Figure 19), turbidity is higher everywhere, and the effect of the 345 bathymety is less evident.

Sandbank-induced high turbidity patterns as in figure 18 are about 2 km wide. Temporal scales of the resuspension-deposition processes are mainly determined by tidal currents. It is therefore not possible to observe these processes with satellite data as they only offer one estimate per day and variations at shorter scales are therefore not measured. Fettweis et al. (2023) showed 350 that in regions with strong tidal regimes, such as in the Belgian coast, the daily sampling from satellites is not enough to capture the variability in the deposition-resuspension cycles caused by tidal currents. Satellite data lack the temporal frequency needed to assess the variability of turbidity at the sandbanks through time, but they provide a relevant tool to analyse the spatial variability at high spatial resolution. Figure 20 shows a Hövmuller diagram of turbidity variations during the year at the same transect shown in figures 18 and 19. In addition to the general trend of decreasing turbidity in the offshore direction, we can 355 appreciate higher turbidity values from January to March. The minimum turbidity is found during June-July in the more off-





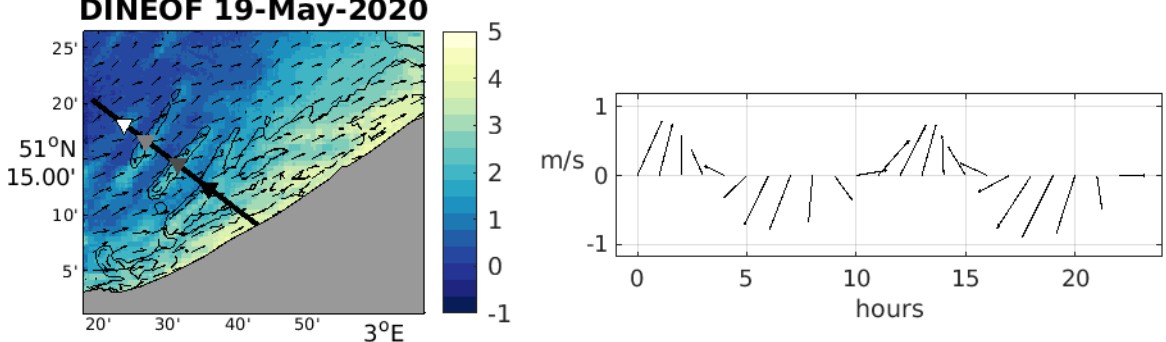

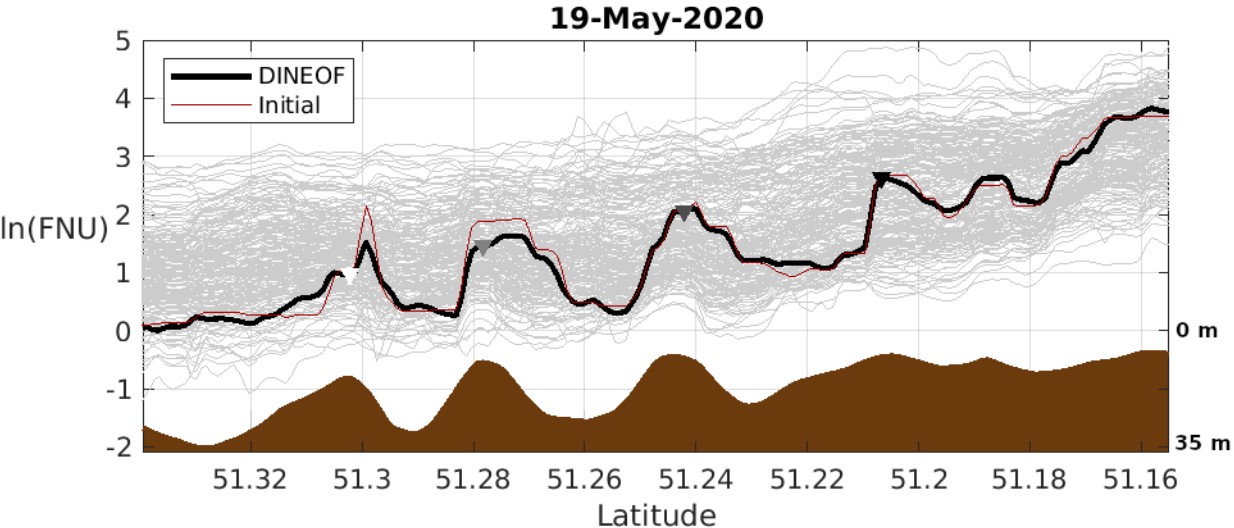

**Figure 18.** Top left panel: DINEOF super-resolution reconstruction of turbidity on 19 May 2020. Black lines show the 5 m and 10 m isobath and the arrows show surface currents at 10AM. The thick black line shows the transect across sandbanks shown in the bottom panel and the triangles are positioned at the top of some of these sandbanks for reference. Top right panel: hourly surface currents during 19 May at the light grey triangle. Bottom panel: across-sandbank transect of turbidity. Light grey lines show all DINEOF 2020 data, the thick black line shows 19 May turbidity, with the triangles shown to ease comparison with the top left panel. The dark red line shows initial turbidity data and bathymetry is shown in brown.



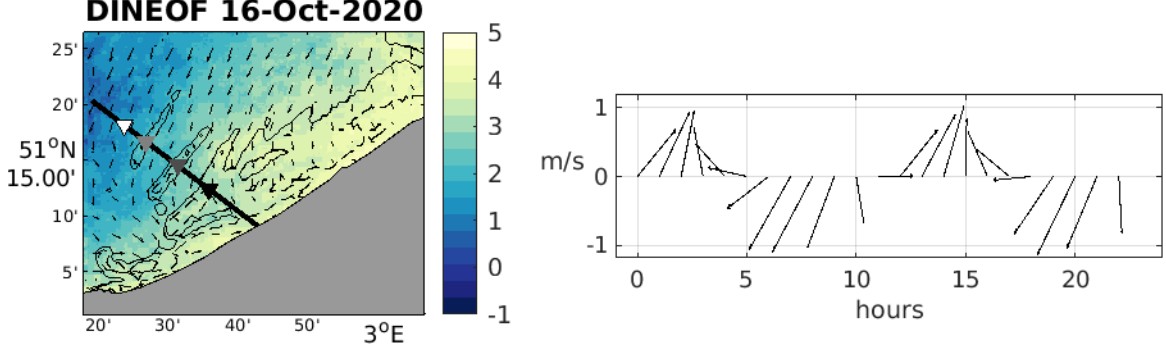

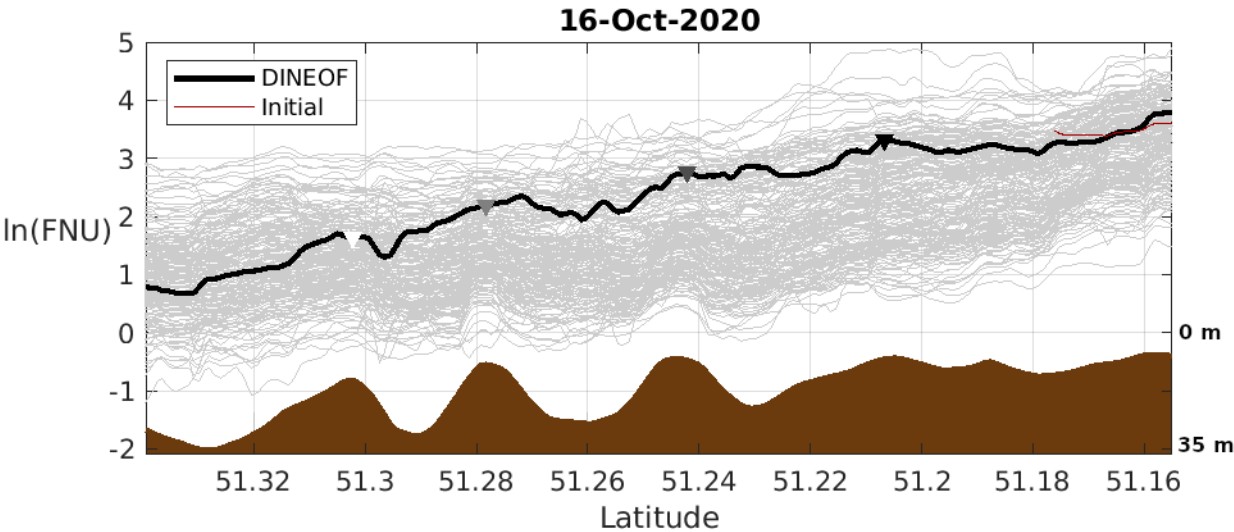

**Figure 19.** Top left panel: DINEOF super-resolution reconstruction of turbidity on 16 October 2020. Black lines show the 5 m and 10 m isobath and the arrows show surface currents at 10AM. The thick black line shows the transect across sandbanks shown in the bottom panel and the triangles are positioned at the top of some of these sandbanks for reference. Top right panel: hourly surface currents during 16 October at the light grey triangle. Bottom panel: across-sandbank transect of turbidity. Light grey lines show all DINEOF 2020 data, the thick black line shows turbidity on 16 October, with the triangles shown to ease comparison with the top left panel. The dark red line shows initial turbidity data and bathymetry is shown in brown.





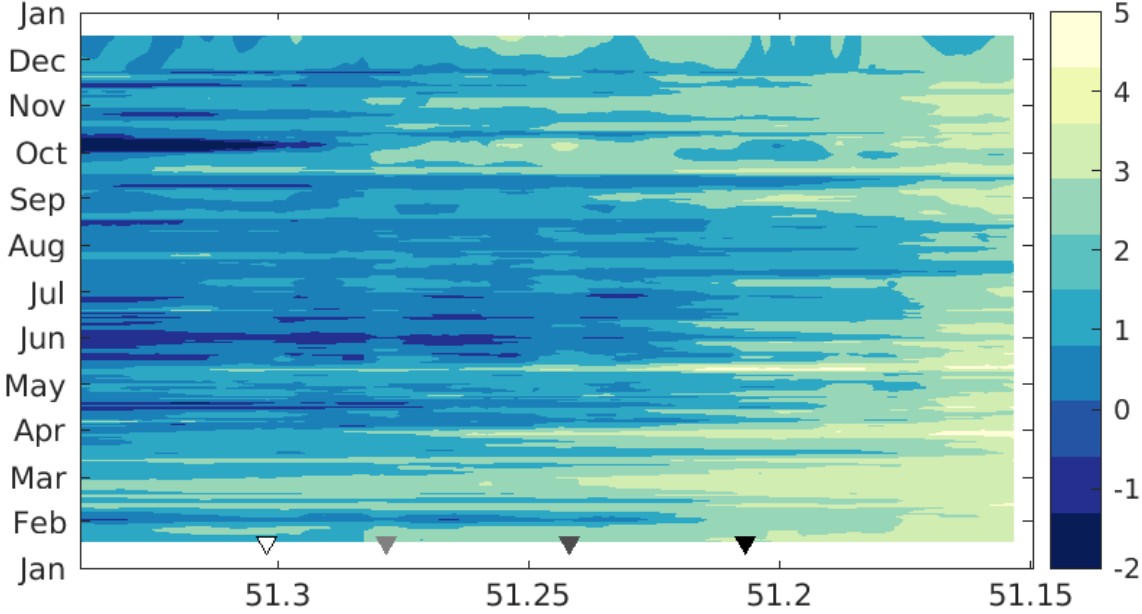

**Figure 20.** Hövmuller diagram showing variations of turbidity at the transect shown in figure 18 from January to December 2020. The position of the sandbanks is marked by triangles at the x-axis, following the same colour code as in previous figures.

shore positions, although there is always a higher turbidity signal at the sandbank positions. Closer to the coast, the minimum in turbidity is found during July-August. Reduced wind-induced mixing can be at the source of this reduction in turbidity. The coastal region is directly influenced by the Scheldt river plume, and this minimum in turbidity is therefore also probably related to a minimum in river outflow during the summer months.


## 6  Conclusions

There are several satellite datasets monitoring ocean colour globally, but each of those has different spatial, temporal and spectral characteristics. It is therefore necessary to develop approaches that allow to use these data streams in a synergistic way. In the case of coastal studies, there is also a need to work at the highest spatial resolution possible, in order to capture the variability that is typical of these regions. All ocean colour satellite sensors are affected by the presence of clouds, and hence these approaches need to also interpolate missing data.

In this work, we have shown an approach to obtain super-resolution cloud-free satellite data using DINEOF. A combination of Sentinel-2 and Sentinel-3 data representing turbidity in the Belgian coast have been used, and the results show that,



working on a combined dataset of Sentinel-2 and Sentinel-3 data, we are able to retain most of the spatial variability present in Sentinel-2 data, and also to increase the spatial variability of the Sentinel-3 data to mimic the Sentinel-2 spatial resolution. The results have been validated using independent in situ data and the ability of DINEOF to increase the spatial resolution has been validated with a chlorophyll dataset covering the whole North Sea. This last example demonstrated that DINEOF is able to recover high spatial resolution information, as compared to the original, high-resolution data that was hidden from the 375 analysis. The approach has been tested in different regions (southern North Sea and Belgian coast in this work) and variables (turbidity and chlorophyll concentration) and can be applied to any other region and variable.

High spatio-temporal resolution data allow to study the small-scale variability in coastal regions, which has been shown in this paper through the influence of sandbanks on turbidity distribution in the Belgian coast. Variables like turbidity or 380 chlorophyll concentration can vary abruptly in a few meters and within a few hours, because of the effect of bathymetry and water currents for example. Using several satellite datasets to analyse these changes allows for a better coverage of the spatio-temporal scales involved.

Super-resolution satellite products obtained from a synergistic use fo several satellite data streams are necessary to study the 385 coastal ocean. The need for high spatial resolution data decreases at omre offshore locations, and therefore the approach presented in this paper could be applied to a multi-resolution dataset with a higher spatial resolution at the most variable regions. Other future developments include the application of the super-resolution DINEOF approach to variables like sea surface temperature, although the absence of high spatial resolution data streams with the necessary accuracy makes this a challenge.

*Code availability.* DINEOF is available at https://github.com/aida-alvera/DINEOF.

*Data availability.* Satellite data used in this work are openly available through the Copernicus Marine Service catalog. This study uses high resolution marine forecast products for the Belgian Coastal Zone as produced by the Royal Belgian Institute of Natural Sciences. The dataset is updated twice a day and can be downloaded at https://erddap.naturalsciences.be/erddap/griddap/BCZ_HydroState_V1.html

*Author contributions.* AAA and DVZ designed the study objectives. AAA implemented the super-resolution DINEOF approach and made 395 the reconstructions. DVD, AD and JM prepared the input datasets and made the validation with in situ data. AB and JMB contributed to the DINEOF implementation and the discussions on the experiments. All authors collaborated on the writing.

*Competing interests.* At least one of the (co-)authors is a member of the editorial board of Ocean Science.



*Acknowledgements.* This work has been carried out as part of the Copernicus Marine Service MultiRes project. Copernicus Marine Service is implemented by Mercator Ocean in the framework of a delegation agreement with the European Union. The Royal Belgian Institute of

Natural Sciences is acknowledged for the ocean currents data.



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
