# Peer review of "Generation of super-resolution gap-free ocean colour satellite products using DINEOF."

_EGUsphere, 2024_

## Author Comment (AC1)

**Reviewer #1**

answers in **_bold italic_**

This is the review of the manuscript "Generation of super-resolution gap-free ocean colour satellite products using DINEOF" by Aida Alvera-Azcárate and co-authors.

The manuscript is written with clear English, addresses a very interesting topic and I surely recommend it to be published in EGUsphere. However, I suggest the authors to go through my comments below which are sincerely meant to improve the quality and readability of the work of which I recognize its scientific value.

At the end of section 3 – after introduction, data and methodology – the reader expects to have a clear idea of what has been done, why and how. Unfortunately this is not the case here and I strongly suggest the authors to reshape the information in a more coherent way to make the reading smoother. For example, lines 198-204 in the results are purely methodological and a reader like myself would likely expect them to appear somewhere in section 3. This surely facilitate going through the paper more easily without the need of reading the entire paper any given time. There are other such examples that prevent the reader to fully get what has been done by the authors: please, carefully address this; the quality of the work will surely improve.

**_We thank the reviewer for their comments. We have tried to streamline the contents to make the reading easier. However, on the example given on lines 198-204, we think that this information, which pertains to this specific implementation and not the overall implementation of the method, should be included when presenting the results, as these choices influence the outcome._**

The second important comment that I would like the authors to take into consideration is given by the fact that results are presented as a series of case studies which on one side do provide the reader with effective and impactful examples but on the other lack of giving a quantitative and statistically robust evaluation of the outcomes of the analyses. I am personally aware that this is not the case but still the authors need to prove it in a more robust and convincing way.

**_We have followed the suggestion by the two reviewers to include percentage change maps in the examples given. The statistically robust evaluation is performed for the validation with in situ data._**

As clearly stated in the acknowledgements, this work was developed in the context of the Copernicus Marine Service MultiRes project, and the way results are presented looks typical of a project final report. Personally, I do not have anything against the large number of figures within a paper; still, twenty figures, for the amount of information they convey, appear definitely too many. Perhaps, authors may consider some of them to be condensed.

**_We have removed figures 8 and 9 from the main paper and put them as supplementary material. Also figure 12, which contained information already present in figure 11, has been completely removed. Figure 20, which showed a Hövmuller diagram along a transect has been also removed._**

Here below the detailed list of comments.

**Section 1 – Introduction**

The problem is well-posed and the line of reasoning clear. I would have expected the authors to choose the area of study in such a way to promote a comparison with the neural network approaches they mentioned in the introduction or at least to use it in one example of the many they show. This would provide robustness to the approach without the need of implementing the other techniques which may require a considerable effort.

***Comparison with other techniques needs to be done with equal datasets, validation settings and validation data, so a direct comparison with any of these approaches would require a new implementation. While not shown on this paper for brevity, we have applied the approach to several regions, obtaining accurate results in all settings.***

Line 24 – the sentence should read (?) "Super-resolution approaches aimed at increasing the spatial resolution of geophysical datasets and have been developed …"

***We have changed the sentence to:***

***"Super-resolution approaches THAT ARE aimed at increasing the spatial resolution of geophysical datasets have been developed"***

***to make it clearer, as it might indeed confuse the reader (the phrasing proposed by the reviewer was not correct).***

**Section 2.1 – Study area**

The area of study is well characterized and provides a useful background for all those unfamiliar with the complex dynamic system of the Belgian waters.

***Thank you***

**Section 2.2 – Satellite data**

This section is a synthetic overview of the not at all trivial preprocessing approach, very useful.

One important information that is missing or too scattered is the space-time resolution of the various datasets that are used in the work along with their temporal coverage. The spatial domain is well depicted by Figure 1. For example, it is not clear why the validation of the Rrs product against in situ data is performed from 2019 to 2022 (not including 2023 and 2024 data) and the generation of the super-resolution data only covers the 2020.

***The application of the DINEOF approach can be done in any period of time. We wanted to use a 1-year time series, and for 2020 we had the largest unbroken timeseries of Hypernet data at the Oostende RT1 station allowing us to validate the super-resolution DINEOF product and its potential in capturing the coastal Turbidity dynamics. We have included this information in section 2.2 to make clear which year we are working with.***

***For the validation of the Remote Sensing product we extended the used Hypernet data to 2019 and 2022 to increase the number of matchups so solidify the assessment. For the years 2019, 2021 and 2022 less Hypernet data was available making it less useful to evaluate the super-resolution product.***

Figure 2 caption – please, substitute "atmospheric correct algorithms" with "atmospheric correction algorithms"

*Done*

**Section 2.2.1 – Remote sensing reflectance and pixel classification**

Since this is a crucial element of the S2 & S3 preprocessing, it would be very useful for the reader if the authors could provide a synthetic overview of the C2RCC to ACOLITE/DSF pixel-based switching which is fully described in Van der Zande et al. (2023).

*We have added the following paragraph to provide an overview of the approach:*

*"To combine the two approaches, a comprehensive, region-independent, and pixel-based automatic switching scheme is required, along with a technique for achieving a seamless transition between the two algorithms. The C2RCC to ACOLITE/DSF pixel-based switching is performed by means of band comparison of the RRS560 and RRS865 products (defined as green-nir ratio) as provided by the C2RCC processor. The green-nir ratio can be modelled using a logarithmic regression curve which starts as linear for the smaller reflectance values, but bends at the point where the saturation of the most sensitive band (i.e. RRS560) occurs. C2RCC pixels which deviate from the logarithmic model are considered erroneous outputs. The ACOLITE/DSF processor has the ability to provide higher RRS ranges compared to C2RCC while being noisier for lower RRS values thus highlighting the complementary between the two approaches. The green-nir ratio value of 45 is selected as the transition point between C2RCC and ACOLITE/DSF products. To ensure a smooth transition between the different ACs, a weighted transition is applied between the green-nir ratio boundaries of 50 and 40 based on the method described by Novoa et al. (2017). The C2RCC to ACOLITE/DSF pixel-based switching is described in detail in Van der Zande et al. (2023). Compatibility between the Sentinel-2/MSI products..."*

Line 106 – from the text it appears that the IDEPIX software is applied soon after the implementation of the two atmospheric correction schemes but from Figure 2 they are at the same level as if the two steps were run simultaneously: it is confusing.

*We changed Line 106 to be less confusing:*

*The IDEPIX software (v2.2.10, algorithm update 8.0.3), available as a The Sentinel Application Platform (SNAP) processor, is used for pixel classification, including cloud masking, cloud shadow identification, sea ice, floating vegetation, sub-pixel objects (ships, small islands and rocks), and the land-water distinction taking temporary water bodies (e.g. intertidal areas, lagoons) into account. SNAP is a software developed by the European Space Agency (ESA) designed for processing and analysing Earth observation data, particularly from the Sentinel satellites. It provides a common architecture for all Sentinel Toolboxes and enables the application of the C2RCC and IDEPIX processor on both Sentinel-2 and Sentinel-3 images.*

In this or in the next section I would have expected to find relevant info about the resolution that is being used for the two sensors.

*We have added this information in section 2.2.2.*

**Section 2.2.2 – Turbidity and Suspended Particulate Matter**

If I understand it correctly, the algorithms used to generate SPM and TUR are an updated version of Nechad et al. (2010) that account for the "switching single band algorithms" developed by Novoa et al (2017). Still the way this paragraph is written is not very clear and I suggest rephrasing it for a better readability.

*We have rephrased this paragraph for better readability*

*The SPM and the TUR products were generated using the generic multi-sensor algorithm described by Nechad et al. (2010). This algorithm provides the theoretical basis for SPM and TUR as a function of reflectance (RRS) at a single band, and provided calibration coefficients for all wavelengths, between 520 nm and 885 nm. It defines a relationship where RRS increases monotonically with SPM/TUR, at first linearly and then tends towards an asymptotic or "saturation" reflectance. This means that RRS becomes insensitive to changes in SPM/TUR which has led to the development of "switching single band algorithms" (Novoa et al., 2017) using using different wavelengths at different SPM concentrations to avoid the saturation effect and typically a smooth weighting between two adjacent spectral bands to avoid image artefacts. The Novoa et al. (2017) approach is applied to both the SPMand TUR products providing a multi-band SPM and TUR product using two bands (red: 665 nm and near-infrared: 865 nm). An example of the TUR products for the Belgian Coastal Zone region is provided in figure 3 showing a good correspondence between both the Sentinel-2/MSI and Sentinel-3/OLCI products providing information at different spatial and temporal resolutions.*

Figure 3 – How do the two maps quantitatively compare? A scatterplot between the two would provide the reader with a better mean to interpret and compare the two products. This figure could then be cited also at lines 103-104 when talking about compatibility between the two sensor products. Please make the numbers on the colorbar larger, they are almost unreadable.

*We have improved the readability of Figure 3 and added a scatter plot to illustrate how both images compare to each other.*

Line 123 – what is the different temporal scale represented by the panels of Figure 3? From the caption it seems that both refer to the 5[th] April, 2020.

*Both panels are on 5 April 2020 as indicated on the caption. This date was chosen to show the same field with the two sensors, to ease the comparison.*

**Section 2.3 – Multi-sensor chlorophyll data**

Line 126 – please spell out CMEMS as Copernicus Marine Environment Monitoring Service (even in braces is fine).

Reading of the text let understand that cmems_obs-oc_atl_bgc-plankton_my_l3-multi-1km_P1D only covers the period February-October 2022. Please, rephrase this sentence.

*Thank you for these two comments, we have changed the text in the manuscript.*

**Section 2.4 – In situ data**

How many in situ-satellite rrs matchups were extracted in the period 2019-2022? How come the 2023 and six months of 2024 were not included in the analysis?

*More information on the number of match ups and total processed satellite images is available in section 4.2 Validation (Line 258-264), e.g. 59 matchups were available for S2/MSI and 179 for S3/OLCI. At the moment of analysis for this manuscript the 2023 and 2024 Hypernet data were*

*not yet available to us. The WaterHypernet systems are still in their development stage with the team working on its quality control and optimization resulting in the fact that the data is not released in NRT mode but rather in batches once the quality can be assured.*

**Section 3.2 – Generation of super-resolution data**

It is not entirely clear why the authors only use data from 2020?

*This is just a choice of period as we could have chosen another one. With DINEOF we do not need long time series to obtain accurate results, the analyses can be done in short-term periods of a few months. As mentioned in a comment above (and included in the manuscript), we have specified in the Data section that 2020 provided the most in situ data for validation, so we chose that period.*

Figure 4 caption – please, substitute spare with square or box.

*Thank you, it has been changed in figures 4 and 5.*

**Section 4.1 – Super-resolution data**

The analysis presented in this section is effective but very qualitative; some hint on how to make it more quantitative at a reasonable cost is provided below.

Lines 202-204 – The only two optional … – what is the range of variability in the results associated with the settings of these two parameters?

*We have added the following text in the paragraph: "As shown in Alvera-Azcárate et al (2009), the use of this filter can result in more EOFs being retained as optimal, which in turn results in a higher variability in the final results. Several tests were performed for values α = 0.01 to α = 0.1 and n = 1 to n = 10 and the combination that maximized the number of EOFs was retained."*

Line 210 – isn't there also a peak in January?

*Light during mid-December to mid-January is very low in the North Sea, which does not allow to accurately derive ocean colour variables. The time series starts therefore on mid-January, and this does not allow to properly resolve the peak that appears to occur in January. We have however mentioned the January peak in the text.*

Line 210 – please substitute "apaprent" with "apparent"

*Done*

Figure 6, 7, 8 – even if these figures do provide a mean to interpret the overall results and the added value of the super-resolution data, this entire analysis lacks of robustness as it only refer to single case studies from which inferring a general rule might be difficult. An important missing information is the data density around the specific examples, that is, how many observations, both high and low resolution, are present in the previous and following days? This, along with the temporal distance between observations and interpolated data, would help explaining where the smaller scale present in the super-resolution data comes from. Probably, a more effective way to evaluate the outcome of the DINEOF interpolation would have been to randomly (or regularly along the time series) remove some day data (both high and low resolution) from the initial time series and use them for a more robust and statistically significant comparison: involving a larger

number of observations.

*It is indeed not easy to assess the 3D results with a handful of figures. However, the examples here are chosen to illustrate what the results are on a few days with different initial settings (Sentinel-2 or Sentinel-3 data, total absence of data, presence of noise…). As DINEOF uses the full 3D matrix to infer the final reconstruction, the presence or not of data just before and after, although very impactful, do not represent the full situation. As for the evaluation of the outcome retaining some information, we tried to do what the reviewer suggests in section "4.3 Scale Assessment", in which initially high resolution data are downgraded and the high resolution is not shown to DINEOF in a setting similar to the Sentinel-3/Sentinel-2 combination done in the present section. This allows to compare the high resolution original data with the DINEOF reconstruction, and the results (especially the zoom shown in figure 14) show that indeed the results have small-scale variability not present in the initial image.*

**Section 4.2 – Validation**

Figures 10, 11 and 12 all have to do with the validation of satellite Rrs (both from S2 and S3) agaist in situ measurements using hypernet data. There is however some inconsistency between the statistics in figures 10 and 11 and those reported in figure 12. Please verify that the numbers in the figures are correct.

**Figures 10 and 11 are consistent and we have omitted figure 12 to reduce the total number of figures in this manuscript.**

Caption of figure 11 – please substitute "Ostend" with "Oostende" or the other way around, consistently with the rest of the manuscript.

*Done*

Figure 12 – I would expect some more degree of spectral consistency between rmse and mape plots, the lack of which could depend on the uneven distribution of the relative error characterized by long tails: perhaps, the median rather than the mean relative error would provide values more in line with respective rmse.

**In order to reduce the number of figures in this manuscript we have omitted figure 12.**

Line 282 – "very well" should be backed up by some statistics.

*In this sentence we use the word "capture" to indicate that we are talking about the qualitative assessment of the results. The statistics are shown in figure 14. We have removed "very well" in the text to avoid confusion.*

Figure 13 – the figure and associated discussion has some potential which should however be supported by some statistics, otherwise the entire paragraph is too qualitative and the reader might see it as only speculative.

To my understanding Figure 13 and Figure 14 contain the same data; figure 13 and associated discussion is too qualitative. On the other hand, Figure 14, if supported by an associated statistics, is

more robust. Perhaps, dots in Figure 14 could be coloured according to time (months or seasons) condensing the information and reducing the number of figures.

*Thank you for this comment. We do think that showing the time series of figure 13 is important, since the improvement in temporal resolution is clearly illustrated. The information on figure 14 is also important, as pointed out by the reviewer, in order to give quantitative metrics about the performance of the results.*

**Section 4.3 – Scale assessment**

Figure 15 – showing the comparison on a selected transect provides a good idea of the outcome of the analysis, which unfortunately falls short of statistical robustness. Perhaps a relative error map instead of the very similar maps (for which it is almost impossible to find differences) would better complement the transect view. This comment also applies to figures 6, 7, 8 (perhaps not crucial because of the cloudiness), 9 and 16.

**The statistical robustness of the results is mostly shown in the validation section. But the reviewer is right that showing change maps can help understand what the technique does. To avoid the large number of figures, we have moved figures 8 and 9 to the supplementary material, and we have added the difference field and the relative percentage change to the two examples of figures 6 and 7.**

**On the day with initially Sentinel-3 data (9 May, figure 6), there are changes observed along the whole domain, with a dominance of along-coast structures following the sandbanks present in the region and that cause changes in turbidity. Most of the changes between the initial and final product affect therefore the structure of these sandbanks. In the day with initially Sentinel-2 data (20 May, figure 7) we observe a similar behaviour, with long and thin structures shown in an along-shore direction, showing that the turbidity around the sandbanks is being modified.**

Lines 318-319 – this sentence is perfectly in line with my previous comment about figure 15. I believe it mostly has to do with the way these results are presented. Another aspect that I would suggest to take into consideration is to try to condense the information as much as possible trying to avoid specific case studies which provide the reader with useful insights but are difficult to be used to derive general rules.

**In that section, 4.3. we show the results in the whole domain to show how they look like overall and then proceed to show a zoom so that the details of the super-resolution are shown. With 3D matrices of more than a hundred time steps, it is difficult to show the details of the results without actually showing a few examples of the final product. The overall quality of the reconstruction is shown in the comparison with in situ data, but to actually show the final details of the super-resolution dataset we think showing the actual data is the best way. The addition of the relative change percentage images does contribute, we hope, to provide a quantification of the results resolution.**

Line 325 – is there any reference or figure to back up this sentence? And perhaps with the drawback of reducing spatial variability by smoothing the field (as mentioned at line 300).

**We have performed a few tests with linearly interpolated data, and they solve the issue of the squares, although we need to assess how this interpolation affects the final variability. We have therefore modified the sentence which now reads:**

**"A linear interpolation could avoid such pattern in the final results, although this would need to be tested."**

**Section 5 – Submesoscale variability in the Belgian Coastal Zone**

This section is a bit controversial: from one side it is presented as the right and expected application for the super-resolution data but on the other it is soon discovered that the data are not suitable to answer the question because of the low temporal resolution. I am not fully sure that this section actually adds value to the work, at least the way it is presented.

Lines 358-359 – even if it is somehow intuitive to assume that during boreal summer river outflows are at their minimum in Europe, it would also be preferable to have a reference to back up this sentence.

**We agree that the lack of high temporal data hinders some of the analyses shown in this section. We have decided to remove figure 20 (the Hövmuller diagram) and related discussion which is the one that indeed would need this higher temporal resolution to be more conclusive. The two examples of specific dates and how the transect changes and is influenced by the presence of the sandbanks are retained, since they are a useful example of how these data can be used to better assess the influence of turbidity at small spatial scales.**

**Section 6 – Conclusions**

Lines 362-366 – as they just mentioned in the previous paragraph, authors should also mention here the importance of the high temporal variability which, unfortunately, is still not covered by the ocean colour sensors currently on orbit.

**We have added this sentence to the end of the third paragraph in the conclusions:**

 **"Satellite data lack however the high temporal resolution that would be needed to study the variability of these small-scale features at adequate temporal scales."**

Line 385 – please substitute "omre" with "more"

*Done*

---

## Author Comment (AC2)

**Reviewer #2**

answers in ***bold italic***

The manuscript by Alvera-Azcárate et al. explores the use of the DINEOF technique on satellite ocean colour data gathered from a coastal area between the North Sea and the English Channel. This technique was applied to achieve a gap-free, interpolated product with enhanced spatial resolution through the combination of Sentinel-2 and Sentinel-3 data. The topic is highly relevant, and the paper could contribute significantly to the field of multi-resolution products. The manuscript is well-organized, reasonably clear, and the quality of the English is good.

However, I have a few suggestions and comments that could improve the clarity and readability of the paper, address some issues/inconsistencies, and should be considered before publication:

**Section 2.2:** It is unclear what the satellite resolutions are and whether Sentinel-3A and Sentinel-3B (or Sentinel-2A and 2B) are treated as merged or separate sensors.

**Lines 101–104:** A brief summary of the switching method employed would be helpful.

**We have added the following paragraph to provide an overview of the approach:**

**"To combine the two approaches, a comprehensive, region-independent, and pixel-based automatic switching scheme is required, along with a technique for achieving a seamless transition between the two algorithms. The C2RCC to ACOLITE/DSF pixel-based switching is performed by means of band comparison of the RRS560 and RRS865 products (defined as green-nir ratio) as provided by the C2RCC processor. The green-nir ratio can be modelled using a logarithmic regression curve which starts as linear for the smaller reflectance values, but bends at the point where the saturation of the most sensitive band (i.e. RRS560) occurs. C2RCC pixels which deviate from the logarithmic model are considered erroneous outputs. The ACOLITE/DSF processor has the ability to provide higher RRS ranges compared to C2RCC while being noisier for lower RRS values thus highlighting the complementary between the two approaches. The green-nir ratio value of 45 is selected as the transition point between C2RCC and ACOLITE/DSF products. To ensure a smooth transition between the different ACs, a weighted transition is applied between the green-nir ratio boundaries of 50 and 40 based on the method described by Novoa et al. (2017). The C2RCC to ACOLITE/DSF pixel-based switching is described in detail in Van der Zande et al. (2023). Compatibility between the Sentinel-2/MSI products..."**

**Figure 2 caption:** Replace "atmospheric correct algorithms" to "atmospheric correction algorithms."

*Done*

**Section 2.3:** Why are multi-sensor satellite data described in a different section from "satellite data"? Aren't they also satellite data? Additionally, at this point in the manuscript, the purpose of the multi-sensor data is unclear, and it is not helpful to have to jump between different pages and sections to understand how these data are being used in the study. I recommend merging this section as a subsection of 2.2 (like the other satellite products) and explaining earlier in the manuscript how these data support the study's goals.

*Thank you for pointing this out, which is indeed a misplaced subsection. It is now section 2.2.3, placed now within the Satellite data section. We have started section 2.2.3 explaining the aim of this dataset:*

*"In order to assess the small-scale information retained in the final DINEOF reconstructions, an additional test on a larger region has been done, using data at 1 km resolution. The aim is to create a degraded 5 km resolution dataset from the initial data and to compare the DINEOF results to the initial, non-degraded dataset. This scale assessment is described in section..."*

4.3. Daily chlorophyll data at a spatial

Please provide a link or reference for the product "cmems_obs-oc_atl_bgc-plankton_my_l3-multi-1km_P1D" mentioned in the paper. This dataset typically covers from September 1997 until 8–10 days before the present day. The authors should clarify why only the data from 1 February 2022 to 1 November 2022 were chosen and why this particular period was selected.

*We have added a link to the DOI of the dataset and an explanation of the choice of period in section 2.2.3:*

*"We have extracted data from 1 February 2022 to 1 November 2022 in order to have a long time series of data, avoiding January and December which have low-light conditions and prevent the calculation of ocean colour variables at the higher latitudes of the domain. The choice of the year was simply to avoid 2020 which is used in the other tests."*

*We have clarified the text and added a link to the data.*

**Lines 149–150:** "The matchup validation protocol described by Bailey and Werdell (2006) was applied to remove erroneous matchups from the analysis." A brief summary of the criteria used to remove erroneous matchups would be useful.

*We have added the following summary:*

*The matchup validation protocol described by Bailey and Werdell (2006) was applied to remove erroneous matchups from the analysis. Macro-pixels of 15x15 60m pixels for Sentinel-2/MSI and 3x3 300m for Sentinel-3/OLCI were extracted from the L2 products. This box allows for the evaluation of spatial stability, or homogeneity, at the validation point. For the satellite data it was required that at least 60% of the pixels in the defined box be valid (i.e. unflagged) to ensure statistical confidence in the mean values retrieved. The arithmetic mean and standard deviation of the non-masked pixels was determined enabling the computation of the coefficient of variation (standard deviation divided by the filtered mean,. Satellite retrievals with extreme variation between pixels in the defined box (CV > 0.15) were excluded from the matchup analysis.*

**Lines 150–151:** "Macro-pixels of 3x3 60m pixels for Sentinel-2/MSI and 3x3 300m for Sentinel-3/OLCI were extracted from the L2 products." Given the different resolutions of these sensors, using the same 3x3 macro-pixel extraction leads to different spatial coverage, affecting spatial variability in macro-pixel extraction, especially in coastal zones. I recommend adjusting the number of pixels for macro-pixel extraction to account for each sensor's resolution to obtain comparable macro-area extractions for matchup analysis.

*We have recalculated the Sentinel-2 matchup statistics for a 15x15 macro pixel to be comparable with the 3x3 macro pixel of Sentinel-3. The match-up metrics do not change significantly*

*compared to the 3x3 macro-pixel for S2 but we agree that this makes it so that the scatterplots for S2 and S3 can be compared directly.*

**Section 3.2:** The authors mention the different spatial resolutions of the satellites, which may cause differences between the datasets. A quantitative analysis (using bias, RMSE, etc.) of these differences, when data from both Sentinel-2 and Sentinel-3 overlap, would be very useful.

*The quantitative difference analysis between both data streams and the reconstructed data are done in section 4.2.*

**Line 197:** Sentinel-2 data are available from 2017 and Sentinel-3 from 2016. Why did the authors only use data from 18 January 2020 to 17 December 2020? Why not a period such as August 2021 to March 2022 or any other period when data from both Sentinel-2 and Sentinel-3 are available?

*For 2020 we had the largest unbroken timeseries of Hypernet data at the Oostende RT1 station allowing us to validate the super-resolution DINEOF product and its potential in capturing the coastal Turbidity dynamics. We have included this information in section 2.2.*

What is the final output resolution? Is it 60m or 100m, as indicated in the caption of Figure 6 (for the first time in the manuscript)?

*It is 60 m, which we considered to be enough for most coastal ocean applications. We have clarified this on the manuscript.*

**Section 4.1**

**line 210:** Regarding Figure 4, I notice a peak in January, which is the highest point in the entire series. Why has this not been considered?

*Light during mid-December to mid-January is very low in the North Sea, which does not allow to accurately derive ocean colour variables. The time series starts therefore on mid-January, and this does not allow to properly resolve the peak that appears to occur in January. We have however mentioned the January peak in the text.*

**Figure 6:** Geographical coordinates in the maps are expressed in degrees and minutes, but in the bottom plot, latitude is in decimal degrees. Consistent units should be used across all figures (this applies to all figures where geographical coordinates are displayed). Also, the bottom plot shows latitude increasing from south to north, while in Figures 18, 19, and 20, latitude increases from north to south. Please ensure consistency in the presentation of geographical coordinates.

*We have changed the axes to show degrees/minutes. On figures 18 and 19 the latitude is reversed so that it is easier to show the correspondence to the above plot. We understand this might be strange, so we have changed the x-axis to show longitude, which increases towards the right.*

**Section 4.2**

**Figures 10 and 11:** It would be helpful to provide definitions and formulas for each of the metrics used.

**Line 263:** The authors refer to band 492, but Figure 10 shows the 490nm plot.

*We have changed the text to match Figure 10*

**Line 267:** The authors mention 179 matchup points for Sentinel-3, but Figure 11 shows matchups ranging between 168 and 179 points. Could this discrepancy be related to the criteria used for match-up quality flagging? Consider eliminating entire spectra when at least one band has quality issues.

*The scatterplots have been redone removing all spectra that get flag in one or more bands*

**Line 270:** The authors refer to bands 492 and 709, but Figure 11 shows 490nm and 704nm plots.

*We have changed figure 11 to match the Sentinel-3 bands and we have changed the text.*

**Line 273:** The authors discuss RMSE for Figure 12, but Figures 10 and 11 show RMSD or cRMSD. Please clarify. Additionally, the metrics in Figure 12 (slope, RMSE/RMSD, and MAPE) differ significantly from those in Figures 10 and 11. Figure 12 should present metrics in a comparable way to demonstrate consistency across the bands. Why are the metrics different?

*Figure 12 indeed was using different metrics as used in figure 10 and 11. Figure 12 was removed since the other reviewer asked to reduce the figures.*

**Figure 13:** The DINEOF (blue line) should have values for every day in 2020. The RT1 (green line) does not have daily values due to quality control or other reasons, but representing it as a continuous line makes it hard to distinguish actual data points. I suggest adding markers on the green line where RT1 measurements exist. I recommend the authors also indicate the number of valid RT1 data points in 2020. Furthermore, why does the green line only span from January to August 2020? Where are the data from September to December 2020, when RT1 data should be available from 2019 to 2023?

*The Panthyr data, represented by the green line actually has multiple data points per day as it collects data every 20 minutes. This means that even after the quality control of the in situ data, it is still expected that the Panthyr RT1 data has the highest temporal frequency, especially compared to the satellite data. Please find the figure with a marker for each actual measurement which makes the plot less readable in our opinion so we suggest to leave it as it is. In total there were 980 observations available from the Panthyr system from January to August. There is no Panthyr data available from September-December as the system was dismounted from the platform for calibration reasons.*

[Figure]

**Lines 286–289:** In this section, just before Figure 14, the authors repeat the technique used for the matchup analysis. However, I assume that the same method was also applied to Figure 13. If this is correct, I recommend moving this paragraph to precede the description of Figure 13. If not, please clarify the method used for the analysis in Figure 13.

*This is indeed the same method used (Bailey and Werdell, 2006). We are keeping the previous order of the text, as changing it reverses the two figures, which in our opinion makes less sense.*

**Figure 14:** Is MAPD the same as MAPE? Also, as described in the text and shown in the figure, with DINEOF the number of matchup points increases from 67 to 90 for 2020. Therefore, can I assume that the maximum number of matchup points available from RT1 is 90? Please clarify this number of in-situ matchup points.

*Yes, MAPD and MAPE are the same metric and we have changed figures 10 and 11 so that we consistently use MAPD throughout the manuscript. You are correct concerning the matchups, the 90 matchups are the result of the quality control where mainly we only allowed matchups with the in situ Panthyr data if the time difference with the satellite observations was smaller than 1 hour. While we have Panthyr data for most days, this does not mean that there is in situ data available within the allowed time difference window due to clouds.*

**Line 294–295:** The authors, referring to Figure 14, mention an underestimation of DINEOF data for high TUR values. Could this underestimation be the same as the one observed in Figure 13 for January and February, possibly due to high cloud cover as indicated in the text a few lines above?

*You are correct, when we replot the data with colours depending on the month of observation you can see that the underestimation mainly comes from the January-February period. We have mention this in the text.*

[Figure]

**Section 4.3:** I assume that the plots in Figures 15, 16, and 17 are in log scale, but this isn't explicitly mentioned. Since the authors note that "the spatial distribution of chlorophyll is similar in all figures," it would be helpful to include percentage difference maps for each analyzed day between "DINEOF Super Resolution" and "DINEOF reference." Additionally, a performance analysis of "DINEOF Super Resolution" over the entire dataset adopted (February 2022–November 2022) using percentage difference maps, scatterplots, or density plots would provide a more comprehensive evaluation than focusing on just 2–3 single days. Expanding the dataset used for analysis is recommended to ensure a more robust evaluation.

***The log units are mentioned in the caption of the figures. We have added a panel with percentage difference map for each of the examples. Doing a percentage difference overall, on the average for example, would not illustrate the detail that are gained through the DINEOF analysis. The evaluation is performed in the Validation section, in which all matchup data are used.***

**Section 5:** In my opinion, the usefulness of this analysis is unclear. While the authors attempt to apply super-resolution interpolated data, this example may not be the most suitable due to high temporal variability, as acknowledged by the authors. It would be more useful if the authors could present the performance of the super-resolution interpolated data over all 210 days selected for 2020. This would help validate the positive results observed for the 2–3 days analyzed in the previous sections and demonstrate the potential utility of this technique for operational contexts or long-term application. Anyway, going into the paragraph, it is not mentioned that the data in all the figures are logarithmic. Moreover, including all the DINEOF data in the bottom panels of Figures 18 and 19 seems unnecessary, as superimposing data from other days makes it harder to interpret the trends for the specific day being analysed.

***We have removed figure 20, which showed a Hövmuller diagram that did not add a lot to the discussion because the data lack the necessary high temporal resolution needed for that. The performance of the 120 days is difficult to show, as any averaging or aggregation of the data does not show the actual resolution of the data. We still believe this section shows how variable this region is and what the influence of sandbanks is on the turbidity.***

---

## Author Response (AR2)

**Reviewer #1**

Thank you to the authors for addressing most of the comments from the first review cycle. I believe the paper is now clearer and more complete.

There are just a few minor corrections left before publication:

**We thank the reviewer for these comments. We have modified the manuscript according to their comments, and we indicate our individual answers below each comment, in bold:**

• Line 131: The word "using" is repeated.

**Thank you, it has been removed**

• Line 310: The authors refer to band 709, but Figure 9 displays a plot for 704 nm.

**Figure 9 has been adapted so that it states the correct band 709nm for Sentinel-3**

• As noted in the initial review, Figure 8 and Figure 9 show a different number of matchups between the bands (53-59 for Figure 8 and 168-179 for Figure 9). In their response, the authors stated: "The scatterplots have been redone removing all spectra that get flagged in one or more bands," but this does not appear to have been done. Please redo the plots by eliminating entire spectra where at least one band has quality issues, ensuring that the number of matchups is consistent across all bands.

**Our apologies for the mix up, now the correct figures 8 and 9 are used with 53 and 168 matchups respectively after removing spectra with a missing band**

**Reviewer #3**

One major concern within the scientific community regarding the use of DINEOF reconstructed data lies in the uncertainty induced by the interpolation process. Review 2 raised a similar concern about the validation part of this work. Personally, I think the authors' response to the previous reviewers' comments are adequate. Considering the substantial effort required for comprehensive validation, their approach is acceptable within the scope of this study. Having said that, I think the paper's discussion needs to be updated to explicitly acknowledge the necessity for further validation. The errors in DINEOF-interpolated results are influenced by both the amount and spatial distribution of missing data (Zhao et al., 2024). Consequently, significant uncertainty may arise in areas with insufficient observations. Adding at least 1-2 sentences to highlight this issue would be necessary especially when discussing results from daily images with a high percentage of missing data.

It was also noted that the authors have cited Becker's paper but did not mention that it provides a method for estimating the error map associated with the reconstruction. While adding this kind of error map may be overly demanding for this study, referencing this method could offer readers a useful resource for estimating errors in DINEOF-interpolated results.

Lastly, I recommend that the authors continue to refine their writing to improve clarity and readability before publication.

**We thank the reviewer for these comments. We have modified the manuscript according to their comments, and we indicate our individual answers below each comment, in bold:**

Reference:

Beckers, J. M., Barth, A., & Alvera-Azcárate, A. (2006). DINEOF reconstruction of clouded images including error maps–application to the Sea-Surface Temperature around Corsican Island. Ocean Science, 2(2), 183-199.

Zhao, H., Matsuoka, A., Manizza, M., & Winter, A. (2024). DINEOF interpolation of global ocean color data: error analysis and masking. Journal of Atmospheric and Oceanic Technology, 41(10), 953-968.

Below are some specific comments:
Line 133: 'SPMand TUR' -> SPM and TUR.
**done**
Line 165: The phrase 'Shifted locations' is somewhat unclear. I suggest specifying coordinates for clarity.
**We have added the shifted coordinates to L165:**
*Matchups for PANTHYR stations were extracted from locations slightly to the East (RT1_shifted, 51.24643°N, 2.92060°E) near the deployment tower, to avoid platform effects such as direct pixel contamination and shadows, as well as in-water wakes.*

Line 198 – 202: Consider rephrasing these sentences for improved readability. Additionally, please clarify the method used for interpolating Sentinel-3 data onto the Sentinel-2 grid—was it nearest-neighbor interpolation? Adding an explanation would be helpful.
**You are correct, we have added the used method and slightly rephrased.**

*"The interpolation of Sentinel-3 data onto the Sentinel-2 grid **using the nearest neighbour method** is done to preserve the size of the matrix, which has to be constant in order to be used in DINEOF, and also to determine the spatial resolution of the final dataset, but no gain in resolution is done at this step."*

Line 216: This sentence is unclear. Please revise for clarity.
**We have rewritten it as follows:**
**"As DINEOF does not re-grid the data, it is important that the initial data are already grided to the final grid that we want to obtain."**

Figure 6: Does the black vertical line represent the transect? If so, why is the line width inconsistent? A similar issue exists in Figure 7.
**Yes this line shows the position of the transect. I think the width change is a plotting artefact. We have provided a version of these two figures with an improved line.**

Figure 6 and Figure 7, bottom panels: Isn't DINEOF only used to reconstruct areas with missing data? If that is the case, what reconstructed data is being used here to calculate the difference map?

**No, DINEOF is used to reconstruct all data, also initially present data. This provides an improved estimate in case of presence of noise and outliers in the initial data, and also avoids sharp gradients at the cloud edges between reconstructed data are initial data. Part of the initial variability can be lost but the final product looks more consistent.**

Line 279 – 285: On days with almost no data, DINEOF can still provide reconstructions with good spatial variability. However, it is important to acknowledge that the accuracy of these reconstructions depends on both the amount and spatial distribution of the missing. On days with large amounts of missing data, the errors in the reconstructed results may be more significant, especially in areas with insufficient observations. Adding 1-2 sentences to discuss this issue would strengthen the discussion, particularly concerning the performance of DINEOF on days with high missing data percentages.
**We have added a sentence explaining this, including a reference to the error estimation by Beckers et al (2006) as requested above:**
**"The accuracy of the reconstruction can be however affected when persistent clouds obscure a specific region for several days (e.g. Alvera-Azcárate et al., 2005; Zhao et al., 2024). The method proposed by Beckers et al.(2006) would allow to obtain a pixel-by-pixel estimation of**

**the reconstruction error variance and can be used to assess the influence of persistent cloud cover on the final result."**

Line: 321 – 325: This section does not align with Lines 165–175, where the authors explained the use of a 2-hour criterion. If this information is redundant, the authors could remove these sentences to avoid repetition.
**We indeed allowed for a 2h time difference between satellite and in situ observations so this sentence was removed as it is redundant:**
***"For the match-up extraction, a maximum time difference of 1 hour between in situ observation and satellite overpass was allowed."***

Line 339: 'Intercalate' or 'interpolate'?
**It is "intercalate". As we did with the previous run, we keep images at high resolution and add images at low resolution at some dates (in this case mimicking the Sentinel-2 Sentinel-3 dates).**

Figure 10: Clarify the temporal resolution and period of the RT1 measurements. For example, does the data end in August 2020?
**This would indeed be useful information, we added this information to L313 and L316:**
*The accuracy of the DINEOF super-resolution products was validated for the Belgian Coastal Zone region by using the hyperspectral in situ data set from the autonomous PANTHYR systems deployed at Research Tower 1 (RT1) near Oostende to generate an in-situ turbidity product which was directly compared with the satellite derived turbidity products. **This resulted in a turbidity time series with a temporal resolution of 20 minutes when daylight was available**.*

*Figure 10 shows the turbidity time series for 2020 overlaying the in-situ data, both the Sentinel-2 and Sentinel-3 turbidity products and the final super-resolution DINEOF gap-filled product showing that the DINEOF product is able to capture the in situ turbidity signal between March and September. In January and February the DINEOF product shows slightly lower values which can be caused by the fact that in  those months the availability of cloud-free satellite products from Sentinel-2 and Sentinel-3 is very scarce. **In situ observations for the period September-December where unavailable as the PANTHYR system was taken down for maintenance.***

Figure 12: Line 346 mentions an initial spatial resolution of 5 km, but the caption refers to 1 km. Please correct this inconsistency. The same issue appears in Figure 14. Ensure that the resolutions of the initial, reference, and DINEOF-estimated data are specified and consistent throughout the text and figures.
**Thank you for spotting this, it has been corrected.**